# LumiNet: Perception-Driven Knowledge Distillation via Statistical Logit Calibration

## Abstract

In knowledge distillation literature, feature-based methods have dominated due to their ability to effectively tap into extensive teacher models. In contrast, logit-based approaches, which aim to distill 'dark knowledge' from teachers, typically exhibit inferior performance compared to feature-based methods. To bridge this gap, we present LumiNet, a novel knowledge distillation algorithm designed to enhance logit-based distillation. We introduce the concept of 'perception', aiming to calibrate logits based on the model's representation capability. This concept addresses overconfidence issues in logit-based distillation method while also introducing a novel method to distill knowledge from the teacher. It reconstructs the logits of a sample/instances by considering relationships with other samples in the batch. LumiNet excels on benchmarks like CIFAR-100, ImageNet, and MSCOCO, outperforming leading feature-based methods, e.g., compared to KD with ResNet18 and MobileNetV2 on ImageNet, it shows improvements of 1.5% and 2.05%, respectively.

## 1 Introduction

The advancement in deep learning models has undergone significant increases in both complexity and performance. However, this progress brings challenges associated with computational demands and model scalability. To mitigate this, knowledge distillation (KD) has been proposed as an efficient strategy (Hinton et al., 2015) to transfer knowledge from a larger, intricate model (teacher) to a more compact, simpler model (student). The primary objective is to trade off performance and computational efficiency. There are two broad categories of KD: logit and feature-based strategies (Romero et al., 2014; Tian et al., 2020; Tung & Mori, 2019; Yim et al., 2017). The logit-based methods aim to match the output distributions of the teacher and student models (Zhang et al., 2018; Mirzadeh et al., 2020; Zhao et al., 2022). In contrast, feature-based methods are centered on aligning the intermediate layer representations between the two models (Romero et al., 2014). In general, it has been observed that feature-based KD outperforms logit-based KD (Zhao et al., 2022). However, feature-based KD suffers from layer misalignment (Romero et al., 2014) (reducing sample density in this space), privacy concerns (Goodfellow et al., 2015) (intermediate model layers accessible for adversarial attacks revealing training data and posing significant threats), and escalating computational requirements (Vaswani et al., 2017; Zhao et al., 2022) (see Fig. 1). These issues raise questions about its effectiveness, particularly in industrial applications. Similarly, these issues underscore the potential merits of logit-based KD over feature-based KD. This paper aims to enhance the effectiveness of logit-based knowledge distillation by leveraging its underlying strengths.

Several reasons underpin the disparity between logit- and feature-based KD. **Firstly**, one significant challenge in logit-based distillation, including data distillation(Zhu et al., 2023), is the issue of overconfidence. This is a common feature of any high-capacity pre-trained teacher model, which tends to assign the highest probability to the target class and shows a high variance in the probability distribution. The primary objective of traditional KD, particularly for the lower-capacity student model, is not just to match the target class probabilities but also to extract the 'dark knowledge' from the teacher model. The 'dark knowledge' refers to the knowledge encoded by the relative probabilities of the non-target classes (Yang et al., 2018; Furlanello et al., 2018; Hinton et al., 2015). However, a key issue is a substantial difference in confidence levels between target and non-target classes, as evidenced in Fig. 1. Even though temperature scaling (Hinton et al.,

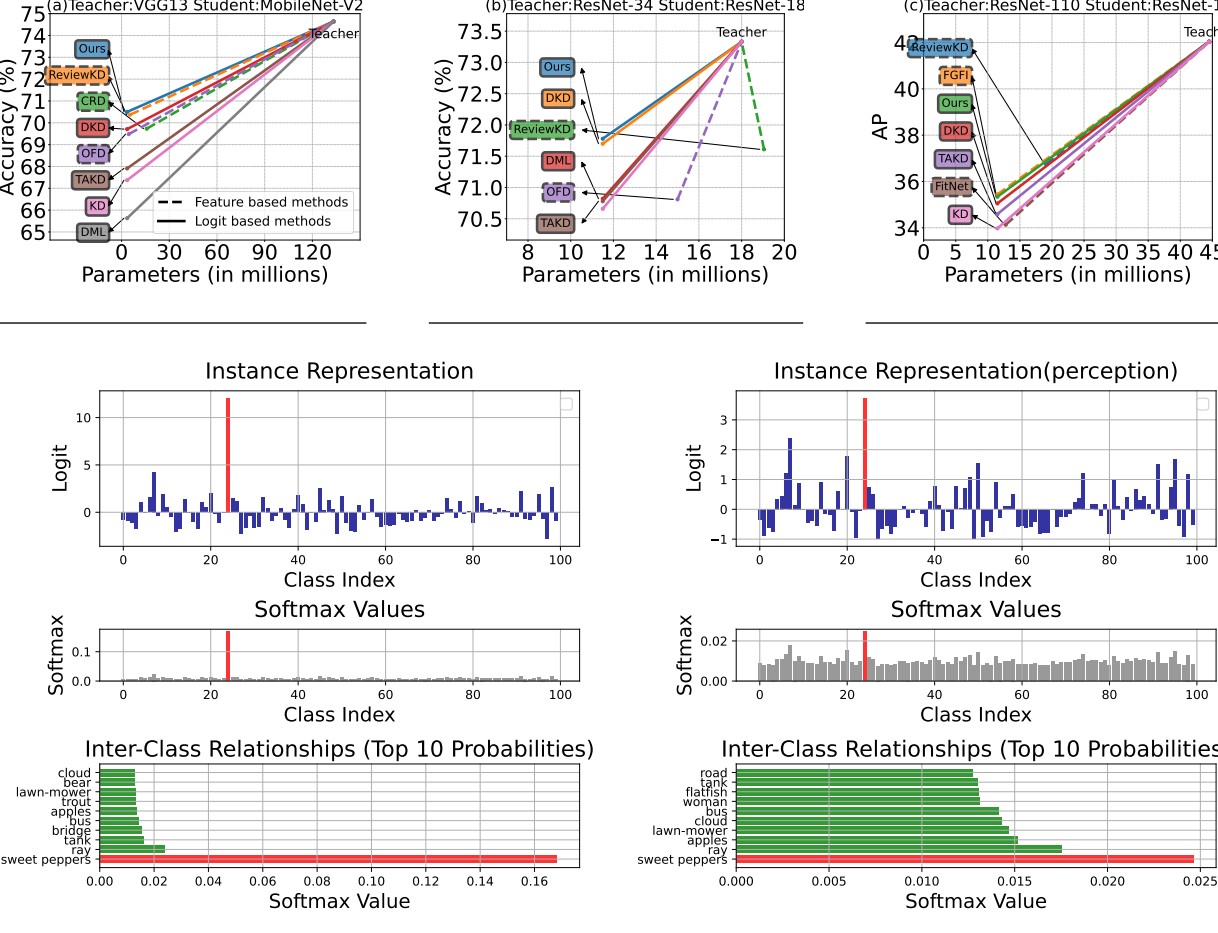

Figure 1: Performance comparison of feature-based and logit-based methods on **(a)** CIFAR-100, **(b)** ImageNet, and **(c)** MS COCO datasets. Our proposed LumiNet, a logit-based method, achieves high accuracy without using extra parameters. **(d-e):** An example of *(Left)* before and *(Right)* after applying our proposed concept perception' on teacher's predicted probabilities. The first set of plots in the top row shows spikes in raw logits for the targeted class (Sweet Peppers, represented in red). This representation changes after the application of perception. Notably, various classes exhibit similar magnitudes to the targeted class, indicating reduced specificity. In the second set, despite conventional knowledge distillation softening, as seen in the left figure, there's persistent overconfidence in the target class. Perception minimizes the difference in softmax values between targeted and non-targeted classes, as depicted in the right figure. The third set illustrates inter-class relationships among the top 10 classes. While conventional scenarios (left) maintain these relationships, the perception method significantly alters them (right).

2015) is employed to address this, determining the optimal value to achieve proper alignment in learning remains an issue (Kim et al., 2021; Chen et al., 2021a; Wang & Yoon, 2021). Moreover, previous attempts to address the issue have led to increased complexity and computational load by introducing multiple objective functions(Zhao et al., 2022; Jin et al., 2023). **Secondly**, most of the logit-based methods often employ a simplistic matching criterion, which might not be robust enough to handle complex logit distributions for the student model, leading to suboptimal knowledge transfer (Romero et al., 2014; Chen et al., 2021a; Wang & Yoon, 2021). **Thirdly**, logit-based KD tends to struggle with granularity. Feature-based methods leverage a broader spectrum of the teacher's knowledge by aligning intermediate representations, providing richer information to the student (Heo et al., 2019a; Bengio et al., 2013; Wang & Yoon, 2021). In contrast, logits provide a more condensed representation, which might not always encapsulate the entirety of the teacher's knowledge (Romero et al., 2014). In addition, the student model learns the distribution of each instance independently in logit-based distillation without considering the distribution of other instances. This leads

to teacher-student fooling Ojha et al. (2023), where the student model may replicate errors or biases present in the teacher's logits for individual instances, affecting its overall performance and generalization. These inherent challenges associated with logit-based distillation necessitate the development of a novel method that strikes a balance between simplicity and effectiveness, mitigating overconfidence issues and enhancing sample representation for better knowledge extraction.

In response to the above challenges, we present LumiNet, a novel approach to knowledge distillation algorithms.The key focus of LumiNet is to generate a new representation of instance-level logit distributions. This representation is expected to effectively address all the issues of the knowledge distillation process discussed earlier. To enhance the model's representation, we focus on the statistical characteristics, mean and variance, of the model output. When focusing on a specific class, we analyze a vector field containing logit values for that class across various samples. Within this vector, we calculate deviation scores (original - mean) for logits, aiding in pinpointing anomalies and highest or lowest logit values. Additionally, by computing the variance, we gain insight into the interrelations among class values. Standardizing logits within the vector provides known statistics, ensuring that, for a particular class vector, the mean of logits is zero (with the maximum value receiving the highest positive score and the minimum value getting the least negative score). While these statistics hold consistently within isolated vector fields, applying this approach to all vector fields in the batch enables the formation of a novel inter-class relationship for each sample/instance. The resulting transformation produces a distinctive representation, which we term 'perception', for each sample, capturing essential insights from the collective statistical behaviors of individual class vectors. This approach effectively addresses overconfidence issues and capacity gaps, eliminating the dark knowledge concept, as illustrated in Fig. 1(d-e). By incorporating the internal relations of other samples/instances within the class, each logit value gains contextual insights, mitigating overconfidence and confirmation bias issues and facilitating the extraction of more subtle knowledge, as evidenced by improved performance scores.

Luminet draws inspiration from Kurt Lewin's Field Theory in Gestalt Psychology Lindorfer (2021), which emphasizes that perception is shaped by the overall psychological field or environment surrounding an entity. Lewin's concept suggests that human goals and behaviors are shaped or reshaped by psychological forces—positive forces drive us toward goals, while negative forces push us away from undesired outcomes. Just as humans perceive objects, Luminet adapts this principle to machine learning. Lumnet dynamically adjusts each sample's representation by leveraging interactions within the surrounding field (the field here is the entire batch), where the forces exerted by other samples influence these adjustments. If a sample encounters suboptimal conditions (overconfidence, errors), the representations of neighboring samples collaborate to minimize its degradation, enhancing overall model robustness. The name "perception" reflects this core idea from Kurt Lewin's Field Theory in Gestalt Psychology.

The performance of LumiNet is evaluated on three computer vision tasks: image recognition, object detection, and transfer learning for feature transfer ability. Our empirical evaluations solidify the efficacy of LumiNet: for instance, using ResNet8x4 as a student, we achieved a notable 77. 5% precision and further established benchmark supremacy across tasks on datasets such as CIFAR100, ImageNet, MS-COCO, and TinyImageNet.

Our contributions are as follows:

- We introduce LumiNet, a novel knowledge distillation algorithm that replaces the traditional dark knowledge concept. It addresses issues of overconfidence, capacity gaps, and teacher-student fooling, while incorporating contextual knowledge into logits by creating new representations for samples.

- Through extensive empirical evaluations, we demonstrate that our method consistently enhances performance across diverse datasets (CIFAR100, ImageNet, MS-COCO, and TinyImageNe), deep learning architectures (ResNet, VGG, ShuffleNet, MobileNet, WRN, and Faster-RCNN-FPN), and tasks (recognition, detection, and transfer learning).

## 2 Related Works

**Logit-based KD:** In the domain of KD, logit-based techniques have traditionally emphasized the distillation process utilizing solely the output logits. Historically, the primary focus of research within logit distillation has been developing and refining regularization and optimization strategies rather than exploring novel methodologies. Noteworthy extensions to this conventional framework include the mutual-learning paradigm, frequently referenced as DML (Zhang et al., 2018), and incorporating the teacher assistant module, colloquially termed TAKD (Mirzadeh et al., 2020). Nonetheless, a considerable portion of the existing methodologies remain anchored to the foundational principles of the classical KD paradigm, seldom probing the intricate behaviors and subtleties associated with logits (Zhao et al., 2022). A novel approach to object detection distillation, combining feature-based and logit-based methods with a closed-loop knowledge distillation framework, has demonstrated improved accuracy and robustness compared to existing state-of-the-art techniques (Song et al., 2024). While the versatility of these logit-based methods facilitates their applicability across diverse scenarios, empirical observations suggest that their efficacy often falls short when juxtaposed against feature-level distillation techniques.

**Feature-based KD:** Feature distillation, a knowledge transfer strategy, focuses on utilizing intermediate features to relay knowledge from a teacher model to a student model. State-of-the-art methods have commonly employed this technique, with some working to minimize the divergence between features of the teacher and student models (Heo et al., 2019b;a; Romero et al., 2014). A richer knowledge transfer is facilitated by forcing the student to mimic the teacher at the feature level. Others have extended this approach by distilling input correlations, further enhancing the depth of knowledge transfer (Park et al., 2019; Tian et al., 2020; Tung & Mori, 2019; Chen et al., 2021b). DiffKD (Huang et al., 2024), a novel knowledge distillation method utilizing diffusion models to denoise and align student features with teacher features, has demonstrated state-of-the-art performance across image classification, object detection, and semantic segmentation tasks These methods, though high-performing, struggle with substantial computational demands and potential privacy issues, especially with complex models and large datasets. These challenges not only amplify processing time and costs but can also limit their practical applicability in real-world scenarios. Recognizing these challenges, we turn our attention to logit-based distillation techniques.

**Applications with KD:** Rooted in foundational work by (Hinton et al., 2015) and further enriched by advanced strategies like Attention Transfer (Zagoruyko & Komodakis, 2017), ReviewKd (Chen et al., 2021b), Decoupled KD (Zhao et al., 2022) and other methods (Park et al., 2019; Tian et al., 2020), KD has significantly improved performance in core vision tasks, spanning recognition (Krizhevsky et al., 2012; Simonyan & Zisserman, 2014; He et al., 2016), segmentation(Qin et al., 2021; Liu et al., 2019), and detection (Li et al., 2022a; Yang et al., 2022; Zheng et al., 2023; Xu et al., 2022). Beyond vision, KD has also made notable strides in NLP tasks like machine translation and sentiment analysis (Kim & Rush, 2016; Zhang et al., 2022). KD has proven valuable in addressing broader AI challenges, such as reducing model biases (Hossain et al., 2022; Chai et al., 2022; Zhou et al., 2021; Jung et al., 2021) and strengthening common-sense reasoning (West et al., 2022). We evaluate our method within the realms of image classification and object detection.

## 3 Methodology

### 3.1 Knowledge Distillation Revisited

Consider a set of distinct samples denoted $\mathcal{X} = \{\mathbf{x}_i\}_{i=1}^{n}$, where $\mathbf{x}_i \in \mathbb{R}^m$ and $n$ represent the total number of samples. Given a parametric deep learning model $f_\theta$ with learnable parameters $\theta$, its output for a sample $\mathbf{x}_i$ is defined as $\mathbf{z}_i = f(\mathbf{x}_i)$, where $\mathbf{z}_i \in \mathbb{R}^c$, and $c$ denotes the number of classes within the sample set $\mathcal{X}$. In the context of KD literature, the model's output $\mathbf{z}$ is often referred to as the logit of the model. For brevity, we will omit $\theta$ from the model notation $f$. To provide more context within the realm of knowledge distillation, we designate $f_T$ as the teacher model and $f_S$ as the student model. The fundamental objective of KD is to minimize the divergence between the logits of the student and teacher for each sample in $\mathcal{X}$. This can be expressed mathematically as minimizing the objective, $L_{\text{KD}} = \sum_{\mathbf{x}_i \in \mathcal{X}} \ell\left(f_T(\mathbf{x}_i), f_S(\mathbf{x}_i)\right)$. Here, $\ell(\cdot, \cdot)$ is a loss function that measures the discrepancy between two vectors. For logit-based distillation, the primary objective is to align the softened logits of the student and teacher models. This alignment is quantified using

the Kullback-Leibler (KL) divergence between the softened probabilities of the two models. Formally, the distillation loss, $L_{KD}$, is defined as:

$$L_{KD} = KL \left( \text{Softmax} \left( \frac{f_T(\mathbf{x}_i)}{\tau} \right) || \text{Softmax} \left( \frac{f_S(\mathbf{x}_i)}{\tau} \right) \right) \tag{1}$$

Here, $\tau$ is the temperature parameter that modulates the softmax sharpness.

The primary hurdles with logit-based distillation lie in the fact that any logit vector $\mathbf{z}_i = f(\mathbf{x}_i)$ is considerably more compact than its feature vector counterpart. Another challenge is the tendency for overconfidence, especially when the output $\mathbf{z}_i$ is from a pre-trained teacher model. Overly confident teacher predictions in knowledge distillation pose a critical issue where $P(y = t|x) \to 1$ for target class $t$, consequently forcing $P(y = i|x) \to 0$ for all non-target classes $i$. This extreme probability distribution effectively suppresses the valuable "dark knowledge" encoded in the relative magnitudes of non-target class probabilities, which Hinton et al. (2015) identified as crucial information for student learning. The resulting near-zero probabilities cause gradient signal degradation during KL-divergence minimization, making it challenging for the student to capture the subtle inter-class relationships that contribute to the teacher's generalization capabilities. These aspects make it difficult for the student model to extract the full range of knowledge embedded in the teacher model (Romero et al., 2014). The following section outlines some potential limitations associated with logit-based knowledge distillation.

**(1) Efficiency and Knowledge Representation:** In neural networks, decision-making frequently relies on the probabilities assigned to specific classes for individual instances or samples (Yegnanarayana, 2009). For example, if the probability of an instance $x_i$ belonging to class $c_1$ is close to the probability of it belonging to class $c_2$, we infer that the two classes are likely to represent similar objects or share common features. We heavily rely on this representation in knowledge distillation, and the student model aims to grasp this pattern (Zhao et al., 2022; Zhang et al., 2018; Mirzadeh et al., 2020). To achieve this, leading logit-based distillation methods often employ multiple objective functions to capture essential information (Zhao et al., 2022; Jin et al., 2023). For instance, Decouple Knowledge Distillation (DKD) (Zhao et al., 2022) uses two objective functions to understand the patterns of target and non-targeted classes separately. However, this approach increases the number of hyperparameters and may pose challenges for industrial applications. On the other hand, MLLD (Jin et al., 2023) adopts multi-level distillation, involving multiple objective functions and introducing computational complexities along with additional hyperparameters. This complexity gives rise to considerations about the feasibility of applying these methods to extensive tasks and industrial contexts. The fundamental question arises: Can we formulate a novel instance representation that simplifies instance-level distillation, mirroring the simplicity of traditional KD? If attainable, this prospect could open the door for a new research trajectory in knowledge distillation, as there would be no need to directly extract relative information about classes from the teacher, which is the core philosophy of KD.

**(2) Role of $\tau$:** In knowledge distillation, the temperature scaling softens the outputs of the teacher's model, serving as a regularizer to reduce overfitting. Moreover, by preventing premature over-confidence in predictions, $\tau$ further promotes better generalization and reduces the risk of fitting too closely to training data (Hinton et al., 2015). Because the outputs of the teacher and student model have inherent statistical differences, finding a suitable value for $\tau$ is difficult (Liu et al., 2023b). Usually, KD methods require extensive $\tau$ fine-tuning, leading to additional computational costs (Rafat et al., 2023).

### 3.2 Introducing LumiNet

Based on our previous discussion, the process of knowledge distillation is found to be further enriched for a given instance $\mathbf{x}_i$ when viewed in the context of its batch samples. The overconfidence issues in inter-class relationships inside instances are also mitigated.

Formally, a measure of information $K$ corresponding to $\mathbf{x}_i$ can be obtained as:

$$K(\mathbf{x}_i) \propto \mathcal{D}(\mathbf{x}_i) + \sum_{j \neq i} R(\mathbf{x}_i, \mathbf{x}_j) \tag{2}$$

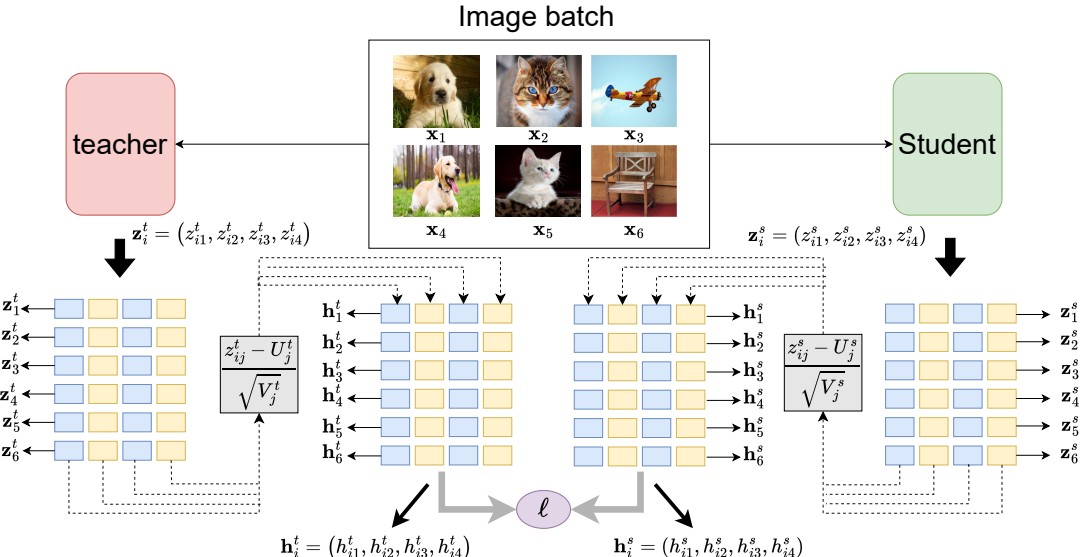

Figure 2: Given a batch of samples $\mathcal{B} = \{\mathbf{x}_1, \mathbf{x}_2, \mathbf{x}_3, \mathbf{x}_4, \mathbf{x}_5, \mathbf{x}_6\}$, both the teacher and student models generate logit for each sample in the batch, denoted as $\mathbf{z}_i^t = (z_{i1}^t, z_{i2}^t, z_{i3}^t, z_{i4}^t)$ and $\mathbf{z}_i^s = (z_{i1}^s, z_{i2}^s, z_{i3}^s, z_{i4}^s)$. In the subsequent stage, $mean((U_j^t, U_j^s))$ and $variance(V_j^t, V_j^s)$ for each class in the batch are computed for both teacher and student logit. These values are then used to normalize the logit of both models, resulting in a new logit representation referred to as the Perception logit: $\mathbf{h}_i^t = (h_{i1}^t, h_{i2}^t, h_{i3}^t, h_{i4}^t)$ and $\mathbf{h}_i^s = (h_{i1}^s, h_{i2}^s, h_{i3}^s, h_{i4}^s)$. Finally, a loss function $\ell$ is calculated between the teacher and student to complete the knowledge distillation process.

Let $K(\mathbf{x}_i)$ be defined as an information measure that quantifies the total information content of an instance $\mathbf{x}_i$. The Relational Measure $R(x_i, x_j)$ quantifies the similarity or mutual information between two instances $x_i$ and $x_j$ in the feature space. Mathematically, it can be expressed as: $R(x_i, x_j) = \text{sim}(f(x_i), f(x_j))$, which is a similarity function such as cosine similarity. The Divergence Measure $D(x_i)$ captures the variance of an instance $x_i$ within its assigned class and is defined as: $D(x_i) = \frac{1}{|C|} \sum_{x_j \in C} \|f(x_i) - f(x_j)\|^2$, where $C$ represents the set of all instances in the class of $x_i$.

**Constructing the perception:** We formulate our approach considering a batch of data samples $\mathcal{B} = \{\mathbf{x}_i\}_{i=1}^{b}$, which is randomly selected from the original dataset $\mathcal{X}$. Consequently, the logits generated by a model $f$ for an instance $\mathbf{x}_i \in \mathcal{B}$ across $c$ classes are represented as: $\mathbf{z}_i = (z_{i1}, z_{i2}, \ldots, z_{ic})$, where $z_{ik}$ symbolizes the logit for the $j^{th}$ class for instance $x_i$. We adjust the logits based on the mean $U_j$ and variance $V_j$ across each class $j$ of a batch. This transformed logit is given by:

$h_{ij} = \frac{z_{ij} - U_j}{\sqrt{V_j}}$. Here, $h_{ij}$ represents the augmented logit for the $j^{th}$ class for instance $\mathbf{x}_i$. Consequently, the augmented logits for instance, $\mathbf{x}_i$ are obtained as:

$$\mathbf{h}_i = \left( \frac{z_{i1} - U_1}{\sqrt{V_1}}, \frac{z_{i2} - U_2}{\sqrt{V_2}}, \ldots, \frac{z_{ic} - U_c}{\sqrt{V_c}} \right) \tag{3}$$

In this context, the reconstructed logits $\mathbf{h}_i$ capture the model's perception. Instead of merely making raw predictions, both the models (teacher and student) try to understand the finer details and differences within the batch of data. As outlined in Eq. 3, the method of constructing 'perceived' logits is explained. In short, when both the teacher and student models' intra-class predictions are adjusted on the same scale, the probability distribution across all the classes for individual instances is influenced. This new set of logits offers us a more insightful representation of each instance. We refer to this set of logits $\mathbf{h}_i$ as 'perception'.

**The LumiNet Loss:** Classical knowledge distillation seeks to transfer the rich perceptual capabilities of a teacher model onto a smaller student model. To this end, we introduce LumiNet, a novel approach

| | Teacher | KD | KD* | Ours |
|---|---|---|---|---|
| Temp | - | 4 | 2 | **4** |
| Entropy | 0.03 | 0.42 | 0.40 | **1.26** |
| Instance Variance | 4.4 | 2.3 | - | **0.91** |
| Mutual Information | 3.64 | 3.60 | 3.56 | **3.65** |
| Avg. Gradient L2 Norm | - | 1.28 | 1.10 | **3.27** |
| Gradient Variance | - | 0.013 | 0.013 | **0.015** |
| Accuracy | 79.42 | 73.08 | 72.91 | **77.50** |

Table 1: Entropy Analysis

| Model | FPR95 (%) ↓ | ECE ↓ | MCE ↓ |
|---|---|---|---|
| | | CIFAR-100 | |
| | | CE / KD / Ours | |
| ResNet8×4 | 3.58 / 4.15 / **2.74** | 0.09 / 0.11 / **0.06** | 0.21 / 0.23 / **0.18** |
| VGG8 | 5.61 / 5.75 / **4.20** | 0.13 / 0.12 / **0.06** | 0.28 / 0.30 / **0.20** |
| MobileNet-V2 | 10.7 / 11.71 / **6.14** | 0.17 / 0.21 / **0.09** | 0.38 / 0.35 / **0.21** |
| WRN-40-1 | 4.13/4.59 / **3.51** | 0.09 / 0.15 / **0.07** | 0.17 / 0.34 / **0.14** |

Table 2: Calibration Analysis

emphasizing the alignment of 'perceptions' rather than raw logits. In LumiNet, we focus on the perceived logits. Given an instance $\mathbf{x}_i$, we denote the logits from the teacher for class $c$ as $h_{ic}^t$ and those from the student as $h_{ic}^s$. The softmax operation scaled by a temperature factor $\tau$ produces probability distributions as: $P_c^T(\mathbf{x}_i) = \frac{\exp(h_{ic}^t/\tau)}{\sum_{c'} \exp(h_{ic'}^t/\tau)}, \quad P_c^S(\mathbf{x}_i) = \frac{\exp(h_{ic}^s/\tau)}{\sum_{c'} \exp(h_{ic'}^s/\tau)}.$

Thus, the LumiNet loss can be represented as:

$$\mathcal{L}_{LumiNet} = \sum_{\mathbf{x}_i \in \mathbf{X}} \sum_c P_c^T(\mathbf{x}_i) \log \frac{P_c^T(\mathbf{x}_i)}{P_c^S(\mathbf{x}_i)}, \tag{4}$$

The objective of the LumiNet loss (Eq. 4) is for the student model to align its 'perception' with the teacher, excluding the direct logit or inter-class relationships. This ensures that the student does not merely imitate the teacher's outputs but learns a deeper understanding of intra-class and inter-class relationships with a new representation (it also aligns with our intent outlined in Eq. 2). By minimizing the LumiNet loss, we ensure that the student model's perception or representation of instances closely mirrors the teacher's, leading to a more robust student model.

**Total Loss Formulation:** The complete training objective for our knowledge distillation framework combines the traditional cross-entropy loss $\mathcal{L}_{CE}$ with our proposed LumiNet loss $\mathcal{L}_{LumiNet}$. While $\mathcal{L}_{CE}$ operates on the raw logits $\mathbf{z}_i$ and ground truth labels $y_i$ to ensure correct classification:

$$\mathcal{L}_{CE} = -\frac{1}{N} \sum_{i=1}^N \sum_{c=1}^C y_{ic} \log(\hat{y}_{ic}) \tag{5}$$

where $(\hat{y}_{ic})$ is the softmax probability, $(y_{ic})$ is the ground truth label, (N) is the batch size, and (C) is the number of classes. The $\mathcal{L}_{LumiNet}$ term works with the perceived logits $\mathbf{h}_i$ to transfer the teacher's perceptual knowledge to the student. The total loss is thus formulated as:

$$\mathcal{L}_{total} = \mathcal{L}_{CE} + \lambda \mathcal{L}_{LumiNet} \tag{6}$$

where $\lambda$ is a balancing scalar that controls the contribution of the LumiNet loss. This dual-objective optimization ensures that the student model not only learns to correctly classify instances through $\mathcal{L}_{CE}$ but also acquires the teacher's rich perceptual understanding through $\mathcal{L}_{LumiNet}$.

### 3.3 Theoretical Foundations and Empirical Validation of LumiNet

**(a) Information-Theoretic Perspective:** The transformation of perception logit, shown in Eq. 3, can be analyzed using mutual information. Let $Z$ be the random variable representing raw logits and $H$ be the perception logits of the same sample. We can show that: $I(H;Y) \geq I(Z;Y)$ where $Y$ is the true class label, and $I(H;Y)$ or $I(Z;Y)$ is mutual information. Where Mmutual information can be defined as: $I(H;Y) = \sum_{h \in H} \sum_{y \in Y} P(h,y) \log \frac{P(h,y)}{P(h)P(y)}$, where $H$ denotes the logits, $Y$ represents the true class labels, and $P(h,y)$ is the joint probability distribution. Also, the inequality holds because the normalization process reduces noise and emphasizes class-relevant information. Our calculations show that our model surpassed the teacher's mutual information score of 3.65.

**(b) Gradient Flow Enhancement:** The gradient of the LumiNet loss (Eq. 4) with respect to the parameters of the student model $\theta_s$ can be expressed as: $\frac{\partial \mathcal{L}LumiNet}{\partial \theta_s} = \frac{\partial \mathcal{L}LumiNet}{\partial h_s} \cdot \frac{\partial h_s}{\partial z_s} \cdot \frac{\partial z_s}{\partial \theta_s}$ The term $\frac{\partial h_s}{\partial z_s} = \frac{1}{\sqrt{V_j}}$

acts as an adaptive learning rate, providing larger updates for classes with lower variance. This theoretically leads to more balanced learning across classes. We tracked the L2 norm of gradients during training for both traditional KD and LumiNet. LumiNet demonstrates a substantially higher average gradient L2 norm (3.27) compared to KD (1.28) and KD* (1.10), indicating stronger parameter updates during training. The gradient 0variance for LumiNet (0.015) is slightly higher than KD and KD *, (also demonstrated in the convergence analysis Fig. 7), suggesting a more dynamic learning process while maintaining overall stability.

**(c) Entropy Analysis:** We measured the entropy of probability distributions derived from raw logits and perception logits across various datasets. The results consistently showed higher entropy for perception logits, as shown in Table 1. This increase in entropy suggests that the perception logits contain more information, supporting our claim in the theoretical analysis. The entropy of LumiNet, which is 1.26, is significantly higher than that of traditional KD (0.42), indicating a higher representation of logits.

**(d) Calibration Analysis:** To evaluate model calibration, we rely on three widely accepted metrics—FPR95 Wei et al. (2022), Expected Calibration Error (ECE) , and Maximum Calibration Error (MCE) Widmann et al. (2019). ECE quantifies the average mismatch between model confidence and accuracy. The confidence range [0,1] is divided into 15 equal-width bins, and ECE is computed as the weighted average of absolute differences between bin confidence and accuracy. MCE identifies the largest such discrepancy across bins (15 bins), reflecting the worst-case calibration error. FPR95 measures the false positive rate when the true positive rate is fixed at 95%. This metric evaluates the reliability of high-confidence predictions and is computed per class in multi-class settings, with the average reported. These metrics comprehensively assess alignment between model confidence and accuracy, highlighting reliability (FPR95), overall calibration (ECE), and worst-case miscalibration scenarios (MCE). We understand overconfidence mitigation by observing improvements in metrics like FPR95, ECE, and MCE, where a lower value indicates fewer false high-confidence predictions and less extreme miscalibration, respectively. These metrics provide a robust framework for analyzing both overconfidence and calibration performance. Our experiments show that our method consistently outperforms models trained using traditional approaches, such as KD or Cross-Entropy (CE) loss, across all three metrics listed in (2). This outcome demonstrates our method's ability to improve calibration while reducing overconfidence in the model's predictions. Additionally, a discussion on the analysis of confirmation bias is included in the Appendix(A6).

### 3.4   The bright side of LumiNet perception

While conventional Knowledge Distillation frameworks face challenges in both logit-based and feature-based implementations, our proposed method sheds light on a renewed perspective to tackle these issues. Here is a detailed exploration of the bright side of LumiNet's approach to KD:

1. **Enhanced logit granularity with perception:** Traditional logit-based approaches are restricted by the inherent granularity of their representations, as characterized by the direct logits of any input $\mathbf{x}_i$. In contrast, LumiNet, leveraging its perception, refines this representation by introducing a transformation. Using the mean $U_j$ and the variance $V_j$ for the logarithms of each class within a batch, as defined in the perceived logits $\mathbf{h}_i$ in Eq. 3, LumiNet achieves a more nuanced understanding. This mathematical recalibration allows the model to encapsulate subtler distinctions and depth, addressing the limitations inherent to conventional logit presentations.

2. **Balanced softening and overfitting:** In traditional KD, the temperature parameter $\tau$ tempers logits by pushing their values closer to zero, effectively reducing variance and bridging the gap between teacher and student logits for efficient knowledge transfer. In LumiNet, logits $\mathbf{x}_i'$ are intra-class normalized, yielding a zero mean and unit variance for each class. Thus, the reliance on $\tau$ for inter-class adjustments is diminished due to the intrinsically reduced variance and mean of the logits.

In essence, LumiNet not only rectifies recognized challenges, but also paves the way to potentially enhance logit-based KD techniques to overshadow their feature-based counterparts.

Table 3: Recognition results on the CIFAR-100 validation.

| | (a) Same architecture | | | | | | (b) Different architecture | | | | |
|---|---|---|---|---|---|---|---|---|---|---|---|
| **Teacher** | ResNet56 | ResNet110 | ResNet32×4 | WRN-40-2 | WRN-40-2 | VGG13 | ResNet32×4 | WRN-40-2 | VGG13 | ResNet50 | ResNet32×4 |
| | 72.34 | 74.31 | 79.42 | 75.61 | 75.61 | 74.64 | 79.42 | 75.61 | 74.64 | 79.34 | 79.42 |
| **Student** | ResNet20 | ResNet32 | ResNet8×4 | WRN-16-2 | WRN-40-1 | VGG8 | ShuffleNet-V1 | ShuffleNet-V1 | MobileNet-V2 | MobileNet-V2 | ShuffleNet-V2 |
| | 69.06 | 71.14 | 72.50 | 73.26 | 71.98 | 70.36 | 70.50 | 70.50 | 64.60 | 64.60 | 71.82 |
| **Feature-Based Methods** | | | | | | | | | | | |
| FitNet (Romero et al., 2014) | 69.21 | 71.06 | 73.50 | 73.58 | 72.24 | 71.02 | 73.59 | 73.73 | 64.14 | 63.16 | 73.54 |
| RKD (Park et al., 2019) | 69.61 | 71.82 | 71.90 | 73.35 | 72.22 | 71.48 | 72.28 | 72.21 | 64.52 | 64.43 | 73.21 |
| CRD (Tian et al., 2020) | 71.16 | 73.48 | 75.51 | 75.48 | 74.14 | 73.94 | 75.11 | 76.05 | 69.73 | 69.11 | 75.65 |
| OFD (Heo et al., 2019a) | 70.98 | 73.23 | 74.95 | 75.24 | 74.33 | 73.95 | 75.98 | 75.85 | 69.48 | 69.04 | 76.82 |
| ReviewKD (Chen et al., 2021b) | 71.89 | 73.89 | 75.63 | 76.12 | 75.09 | 74.84 | 77.45 | 77.14 | 70.37 | 69.89 | 77.78 |
| FCFD(Liu et al., 2023a) | 71.68 | - | 76.80 | 76.34 | 75.43 | 74.86 | 78.12 | 77.81 | 70.67 | 71.07 | 78.20 |
| **Logit-Based Methods** | | | | | | | | | | | |
| KD(Hinton et al., 2015) | 70.66 | 73.08 | 73.33 | 74.92 | 73.54 | 72.98 | 74.07 | 74.83 | 67.37 | 67.35 | 74.45 |
| DML(Zhang et al., 2018) | 69.52 | 72.03 | 72.12 | 73.58 | 72.68 | 71.79 | 72.89 | 72.76 | 65.63 | 65.71 | 73.45 |
| TAKD(Mirzadeh et al., 2020) | 70.83 | 73.37 | 73.81 | 75.12 | 73.78 | 73.23 | 74.53 | 75.34 | 67.91 | 68.02 | 74.82 |
| DKD(Zhao et al., 2022) | 71.97 | 74.11 | 76.32 | 76.24 | 74.81 | 74.68 | 76.45 | 76.70 | 69.71 | 70.35 | 77.07 |
| TTM(Zheng & Yang, 2024) | 71.83 | 73.97 | 76.17 | 76.23 | 74.32 | 74.33 | 74.18 | 75.39 | 68.98 | 69.24 | 76.57 |
| **Ours** | **72.29** | **74.2** | **77.50** | **76.38** | **75.12** | **74.94** | **76.66** | **76.95** | **70.50** | **70.97** | **77.55** |
| | (0.11) | (0.23) | (0.19) | (0.32) | (0.13) | (0.16) | (0.08) | (0.15) | (0.10) | (0.12) | (0.21) |
| Δ | +1.63 | +1.12 | +4.17 | +1.37 | +1.58 | +1.96 | +2.59 | +2.12 | +3.13 | +3.62 | +3.1 |

# 4 Experiments

## 4.1 Setup

**Dataset:** Using benchmark datasets, we conducted experiments on three vision tasks: image classification, object detection, and transfer learning. Our experiments leveraged *four* widely acknowledged benchmark datasets. First, **CIFAR-100** (Krizhevsky et al., 2009), encapsulating a compact yet comprehensive representation of images, comprises 60,000 32x32 resolution images, segregated into 100 classes with 600 images per class. **ImageNet** (Russakovsky et al., 2015), a more extensive dataset, provides a rigorous testing ground with its collection of over a million images distributed across 1,000 diverse classes, often utilized to probe models for robustness and generalization. Concurrently, the **MS COCO** dataset (Lin et al., 2014), renowned for its rich annotations, is pivotal for intricate tasks, facilitating both object detection and segmentation assessments with 330K images, 1.5 million object instances, and 80 object categories. We strictly adhered to standard dataset splits for reproducibility and benchmarking compatibility for training, validation, and testing. The **TinyImageNet**[1] dataset, although more compact, acts as an invaluable resource for transfer learning experiments due to its wide variety across its 200 classes.

**Network architectures:** Various architectures are employed depending on the context. For CIFAR-100, homogeneous configurations use teacher models such as ResNet56, ResNet110 (He et al., 2016), and WRN-40-2, paired with corresponding students such as ResNet20 and WRN-16-2 (Table 3a). In heterogeneous settings, architectures such as ResNet32×4 and VGG13 (Simonyan & Zisserman, 2014) for teachers are paired with lightweight models like ShuffleNet-V1, ShuffleNet-V2 (Ma et al., 2018) and MobileNet-V2 (Sandler et al., 2018) as students (Table 3b). For ImageNet classification, ResNet34 was employed as the teacher and ResNet18 as the student. Additionally, for object detection on MS-COCO, Faster RCNN with FPN (Zhang et al., 2022) was utilized as a feature extractor, with predominant teacher models being ResNet variants, while the latter served as a student. A pre-trained WRN_16_2 model is further harnessed for transfer

---

[1] https://www.kaggle.com/c/tiny-imagenet

Table 4: Reported are the Top-1 and Top-5 accuracy (%) on ImageNet validation.

| | | | Feature-Based Methods | | | | Logit-Based Methods | | | | | |
|---|---|---|---|---|---|---|---|---|---|---|---|---|
| | | | **ResNet34 (Teacher) and ResNet18 (Student)** | | | | | | | | | |
| | Teacher | Student | AT | OFD | CRD | ReviewKD | KD | DML | TAKD | DKD | **Ours** | Δ |
| Top-1 | 73.31 | 69.75 | 70.69 | 70.69 | 70.81 | 71.17 | 70.66 | 70.82 | 70.78 | 71.70 | **72.16** | +1.5 |
| Top-5 | 91.42 | 89.07 | 90.01 | 90.01 | 89.98 | 90.13 | 89.88 | 90.02 | 90.16 | 90.41 | **90.60** | +0.72 |
| | | | **ResNet50 (Teacher) and MobileNet-V2 (Student)** | | | | | | | | | |
| Top-1 | 76.16 | 68.87 | 69.56 | 71.25 | 71.37 | 72.56 | 70.50 | 71.35 | 70.82 | 72.05 | **72.55** | +2.05 |
| Top-5 | 92.86 | 88.76 | 89.33 | 90.34 | 90.41 | 91.00 | 89.80 | 90.31 | 90.01 | 91.05 | **91.12** | +1.32 |

Table 5: Detection results on MS-COCO using Faster-RCNN-FPN (Lin et al., 2017) backbone.

| | | | Feature-Based Methods | | | Logit-Based Methods | | | | |
|---|---|---|---|---|---|---|---|---|---|---|
| | | | **ResNet101 (Teacher) and ResNet18 (Student)** | | | | | | | |
| | Teacher | Student | FitNet | FGFI | ReviewKD | KD | TAKD | DKD | Ours | Δ |
| AP | 42.04 | 33.26 | 34.13 | 35.44 | **36.75** | 33.97 | 34.59 | 35.05 | 35.34 | +1.37 |
| $AP_{50}$ | 62.48 | 53.61 | 54.16 | 55.51 | 56.72 | 54.66 | 55.35 | 56.60 | **56.82** | +2.16 |
| $AP_{75}$ | 45.88 | 35.26 | 36.71 | **38.17** | 34.00 | 36.62 | 37.12 | 37.54 | 37.56 | +0.94 |
| | | | **ResNet50 (Teacher) and MobileNet-V2 (Student)** | | | | | | | |
| AP | 40.22 | 29.47 | 30.20 | 31.16 | **33.71** | 30.13 | 31.26 | 32.34 | 32.38 | +2.25 |
| $AP_{50}$ | 61.02 | 48.87 | 49.80 | 50.68 | 53.15 | 50.28 | 51.03 | 53.77 | **53.84** | +3.56 |
| $AP_{75}$ | 45.88 | 30.90 | 31.69 | 32.92 | 36.13 | 31.35 | 33.46 | 34.01 | 33.57 | +2.22 |

learning. We also performed tests on ViT models (Dosovitskiy et al., 2021). DeiT-Ti (Touvron et al., 2021), PiT-Ti(Heo et al., 2021), PVT-Ti(Wang et al., 2021), and PVTv2-B0(Wang et al., 2022) served as student models, with ResNet50 acting as the teacher model.

**Evaluation metric:** We assess methods' performance using Top-1 and Top-5 accuracy for classification tasks. We employ Average Precision (AP, AP50, and AP70) to gauge precision levels in object detection tasks. We calculate a Δ that denotes the performance improvement of LumiNet over the classical KD method, underlining the enhancements of our approach.

**Implementation details:** We explore knowledge distillation using two configurations: a homogeneous architecture, where both teacher and student models have identical architectural types (ResNet56 and ResNet20), and a heterogeneous architecture, where they differ (ResNet32x4 as the teacher and ShuffleNet-V1 as the student). Our study incorporates a range of neural network architectures such as ResNet, WRN, VGG, ShuffleNet-V1/V2, and MobileNetV2. The training parameters are set as follows: for CIFAR-100, a batch size of 64 and a learning rate of 0.05; for ImageNet, a batch size of 128 and a learning rate of 0.1; and for MS-COCO, a batch size of 8 with a learning rate of 0.01. We followed the implementation settings of (Zhao et al., 2022). To implement distillation in the ViT variant, we adopted the implementation settings detailed by (Li et al., 2022b). All models are trained on a single GPU. Detailed implementation for each task can be found in the appendix.

## 4.2 Main Results

**Comparison methods:** We compare our method with well-established feature- and logit-based distillation methods, underscoring its potential and advantages in the knowledge distillation domain. Notable methods in *Feature Based Methods* category include FitNet (Romero et al., 2014), which aligns features at certain intermediary layers, RKD (Park et al., 2019) that focuses on preserving pairwise relations of examples, and CRD (Tian et al., 2020), which minimizes the contrastive loss between the representations of the teacher and student models. Other methods in this category include OFD (Cho & Hariharan, 2019) and ReviewKD (Chen et al., 2021b), each bringing unique strategies to leverage intermediary network features. *Logit Based Methods* methods include KD (Hinton et al., 2015), DML (Zhang et al., 2018), TAKD (Mirzadeh et al., 2020), and DKD (Zhao et al., 2022), which ensure that the student's logits are similar to the teacher's.

Table 6: Comparison of training time per batch, number of extra parameters ($\theta$) on the CIFAR-100.

| Teacher: ResNet32×4 Student: ResNet8×4 | KD | RKD | FitNet | OFD | CRD | ReviewKd | DkD | **Ours** |
|---|---|---|---|---|---|---|---|---|
| Latency↓ (ms) | 11 | 25 | 14 | 19 | 41 | 26 | 11 | **11** |
| $\theta$ ↓ | 0 | 0 | 16.8K | 86.9K | 12.3M | 1.8M | 0 | **0** |
| Acc↑ (%) | 73.33 | 71.90 | 73.50 | 74.95 | 75.51 | 75.63 | 76.32 | **77.50** |

Table 7: Results after applying Auto Augmentation.

| Teacher | WRN_40_2 | WRN_40_2 | VGG 13 | ResNet32×4 | WRN-40-2 |
|---|---|---|---|---|---|
| Accuracy | 75.61 | 75.61 | 74.64 | 79.42 | 75.61 |
| Student | WRN_16_2 | WRN_40_1 | VGG 8 | ShuffleNet-V2 | ShuffleNet-V1 |
| Accuracy | 73.26 | 71.98 | 70.36 | 71.82 | 70.50 |
| MLLD | 76.63 | 75.35 | 75.18 | 78.44 | 77.44 |
| Ours | **76.91** | **76.01** | **75.57** | **79.12** | **77.97** |
| Δ | +0.28 | +0.66 | +0.39 | +0.68 | +0.53 |

**Recognition tasks:** We perform image recognition tasks on CIFAR-100 and ImageNet. On **CIFAR-100**, when teacher and student models shared identical architectures, shown in Table 3a, LumiNet presented improvements of 2-3%. And when the architectures were from different series, shown in Table 3b, the improvements were between 3-4%, consistently outperforming the baseline, classical KD, and other methods rooted in KD's principles. Similarly, on the intricate **ImageNet** dataset, LumiNet outshined all logit-based distillation techniques and beat state-of-the-art feature-based distillation methods, shown in Table 4. These results consistently demonstrate that, regardless of variations in the dataset or architectural differences, LumiNet performs exceptionally well. In particular, it highlights the distinctive ability of LumiNet to learn based on the concept of 'perception.

In Table 6, we show that LumiNet demonstrates a superior trade-off between the number of extra parameters/running time vs. precision. The necessity for extra parameters in feature-based techniques arises from integrating projection or intermediate layers that align the teacher's feature space to the student model. With a latency of 11 ms, our method matched the best-performing models in speed and is exceptionally efficient (77.50%) without extra parameters. This combination of low latency, less computation, and high accuracy further underscores the exceptional effectiveness and efficiency of LumiNet.

**Detection task:** The quality of deep features is crucial for accurate object detection. One persistent challenge is effective knowledge transfer between established teacher models and student detectors (Li et al., 2017). Generally, logits cannot provide knowledge for object localization (Wang et al., 2019). Although logit-based techniques have traditionally been used for this, they often do not meet state-of-the-art standards. On **MS COCO** dataset, LumiNet delivered noticeably better results (Table 10) compared to logit-based methods, which are comparable to feature-based methods. Also, it is possible to enhance accuracy through hyperparameter tuning. Additionally, we enhance our approach by integrating a feature-based technique. The combination of these two methods yields state-of-the-art results, as detailed in the appendix.

**Transfer learning task:** To assess the transferability of deep features, we carry out experiments to verify the superior generalization capabilities of our algorithm LumiNet. In this context, we used the Wide Residual Network (WRN-16-2), distilled from WRN-40-2, as our principal feature extraction apparatus. Subsequently, sequential linear probing tasks were performed on the benchmark downstream dataset, notably *Tiny-ImageNet*. Our empirical results, delineated in Fig. 2(a), manifestly underscore the exemplary transferability of features cultivated through LumiNet.

**Effect of Strong Augmentation:** In Table 7, we report performance after using auto-augmentation by increasing the complexity of training samples (Cubuk et al., 2019). LumiNet outperforms auto augmentation-based method (Jin et al., 2023) in heterogeneous and homogeneous settings on the CIFAR-100 dataset. The results show our effectiveness in distilling knowledge from challenging samples.

Table 8: Top-1 mean accuracy (%) comparison on CIFAR-100

| Student | Vanilla | KD | AT | SP | LG | AutoKD | Ours | $\Delta$ |
|---------|---------|----|----|----|----|--------|------|----------|
| DeiT-Ti | 65.08 | 73.25 | 73.51 | 67.36 | 78.15 | 78.58 | **79.05** | +5.8 |
| PiT-Ti | 73.58 | 75.47 | 76.03 | 74.97 | 78.48 | 78.51 | **79.80** | +4.33 |
| PVT-Ti | 69.22 | 73.60 | 74.66 | 70.48 | 77.07 | 77.48 | **78.12** | +4.52 |
| PVTv2-B0 | 77.44 | 78.81 | 78.64 | 78.33 | 79.30 | 79.37 | **79.94** | +1.13 |

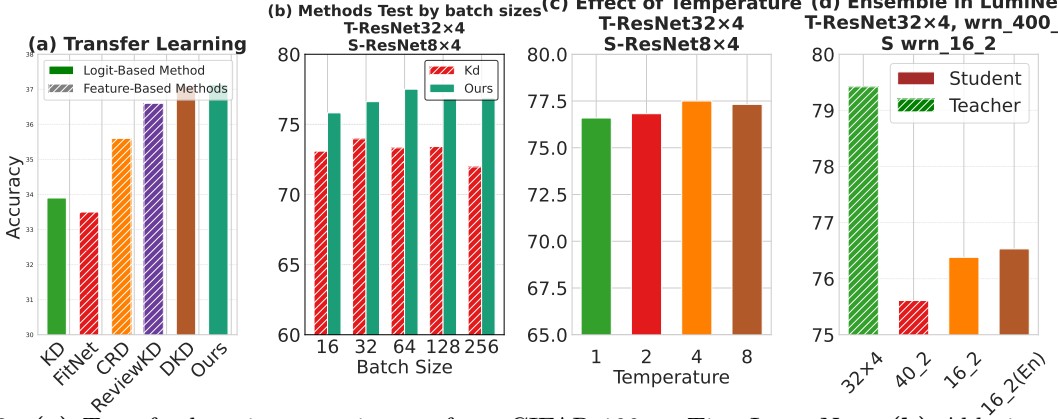

Figure 3: **(a)** Transfer learning experiments from CIFAR-100 to Tiny-ImageNet. **(b)** Ablation study on different batch sizes. **(c)** Impact of different $\tau$ values. **(d)** Performance on ensemble learning.

**Vision Transformer:** To explore the capabilities of LumiNet beyond conventional ConvNet models, we performed experiments using different variants of vision transformers (ViT) in the CIFAR-100 dataset. We trained ViT with the optimal distiller obtained using ResNet-56 as a CNN teacher. Table 8 presents the results of experiments that involve both vanilla and distillation models in a variety of distillation methods. The results indicate a notable improvement in the performance of vision transformers with the application of LumiNet, showcasing improvements ranging from 2% to 14% compared to vanilla. In particular, LumiNet consistently outperforms other methods, demonstrating improvements of 1 to 6% compared to KD, particularly. The effectiveness of our proposed method is affirmed in various ViT architectures, highlighting its versatility. It is essential to emphasize that our approach, despite being a straightforward logit-based( soft logits) method in this context, proves to be more effective in transformer-based architectures compared to feature-based distillation methods.

### 4.3 Ablation study

**Varying batch sizes:** Fig. 3(b) showcases an ablation study that compares the performance of the LumiNet method with both a basic student model and the KD method in various batch sizes. Batch sizes range from 32 to 256. The student model, which serves as a standard baseline, demonstrates a slight decline in performance as the batch size increases. In comparison, LumiNet consistently outperforms both the student and the KD methods in all batch sizes tested, suggesting its robustness and superiority in the given context.

**Varying $\tau$:** The logits within our perception framework are reconstructed with a clear statistical understanding of intra-class logits. For this, both the teacher and the student models exhibit "softened" values, achieved through normalization by variance and maintaining an intra-class mean of zero. Consequently, the dependency on temperature $\tau$ is minimal. Empirical evaluations in Fig. 3(c) suggest minimal performance fluctuations across $\tau$ (ranging between 1 and 8) yield optimal results.

**Ensemble of teachers:** We employ an ensemble of two teacher models: ResNet 8x4 and WRN-40-2 (labeled in the figure as "8x4" and "40-2"). This ensemble technique, which we term "Logit Averaging Ensemble," involves averaging the logits produced by the two teacher models (Sagi & Rokach, 2018). When training the student model, WRN-16-2 (labeled as "16-2" for the regular student and "16-2(en)" for the student

learned by ensemble technique), we observed a notable improvement in accuracy using this ensemble-derived guidance. As shown in Fig. 3(d), when conventionally training with our LumiNet approach with only the WRN-40-2 teacher, we achieve an accuracy of 76. 38%. However, the results improve slightly to 76. 52% when training is augmented with insights from the ensemble technique. This suggests that the ensemble's aggregated information potentially enables the student model to capture more intricate patterns and nuances from the teachers.

### 4.4 Discussion

**Achievments of LumiNet:** LumiNet addresses a fundamental challenge in knowledge distillation. The inherent capacity gap between teacher and student models manifests in their logit distributions. Traditional methods often struggle when student models, with their limited capacity, attempt to match the complex, high-variance logit distributions of teacher models. This perception-based calibration simultaneously normalizes both teacher and student logits, significantly reducing their distributional variance and eliminating teacher overconfidence. This dual calibration effectively narrows the representational gap between models, making it easier for the student to converge to optimal solutions despite its capacity constraints. The empirical evidence supports this through improved gradient flow (3.27 L2 norm compared to KD's 1.28) and higher entropy scores (1.26 versus 0.42), demonstrating that when both teacher and student operate in a more balanced, normalized logit space, the student can more effectively learn and generalize despite its architectural limitations. The experiments show that LumiNet outperforms numerous benchmark datasets, including CIFAR-100, ImageNet, and MS-COCO. We observed accuracy gains ranging from 1.5% to 4.17% over classic distillation methods, without additional parameters in training settings.

In addition, it likely solves another known problem in KD. When kD is performed per instance, the student model risks inheriting errors or biases from the teacher, a problem known as teacher-student fooling(Ojha et al., 2023). LumiNet mitigates this by incorporating relationships across all instances in a batch and recalibrating logits using batch-level statistics. With batches randomly sampled and later recalibrated the logits space by considering other samples in the batch, no single instance disproportionately influences the student, reducing the chance of transferring biases or errors. This ensures a more robust and generalized student model, resilient to dataset imperfections and teacher model flaws.

**Limiation:** LumiNet has some drawbacks in spite of its advantages. In complex or multi-modal tasks where intermediate feature representations are essential, it might not perform likewise. The performance of the model with small or less diverse batches may be limited by its dependence on batch-level relationships. Although LumiNet has demonstrated impressive performance in computer vision tasks, it is still unclear if it can be applied to non-visual fields like natural language processing.

**Future work:** Future research could examine LumiNet's approach to KD outside of computer vision, as it is thought to have significant potential in other disciplines. The perception-based logit calibration technique could be used to improve the deployment and compression of large language models in resource-constrained environments. Furthermore, LumiNet could be used for continual learning settings to investigate how the method can aid in successful knowledge acquisition while avoiding catastrophic forgetting.

## 5 Conclusion

We propose LumiNet, a novel knowledge distillation method, which introduces a unique representation for instances through a concept we term 'perception.'In this novel representation, we depart from the fundamental philosophy of classical KD, which centers around extracting relative information from the teacher model. Within this framework, our main focus lies on addressing overconfidence issues to achieve improved optimization. It also tackles the capacity gap issue, where the student model struggles to learn due to the high variance in the teacher model's logit distribution. In addition, we integrate statistical knowledge from other instances into an instance, resulting in a substantial improvement in accuracy compared to leading methods, which mitigates the problem of teacher-student fooling, where the student model can be misled by the teacher when it only relies on sole instance distributions. Also, LumiNet demonstrates efficiency on par with traditional KD, solidifying its suitability for industry adoption. Our comprehensive empirical experi-

ments, spanning recognition using both convnets and vision transformers, detection, and transfer learning, consistently highlight the superior performance of LumiNet.

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

## A  Appendix

### A.1  Alternative Perspective on Traditional Knowledge Distillation Method.

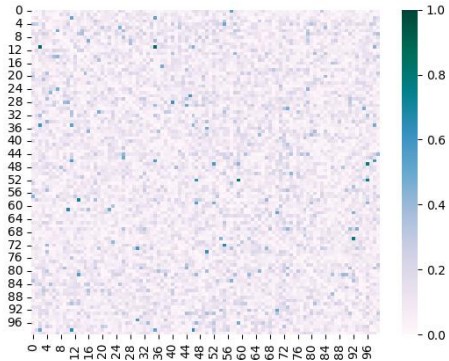

Figure 4: Comparing teacher-student prediction similarity in DKD method.

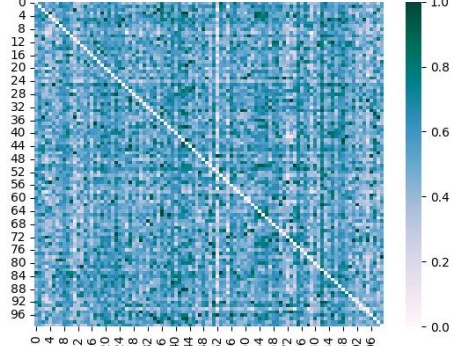

Figure 5: Comparing Teacher-Student Predictions Similarity in Our Method

The primary goal of traditional knowledge distillation is to replicate the raw logits of the teacher, as illustrated in Fig 4. This figure demonstrates that the predictions closely resemble the teacher's logits.In this method, we often face overconfidence issues, resulting in inferior performance compared to feature-based KD. Moreover, despite our aim to mimic the logits of the teacher, a substantial gap persists between teachers and students. However, in our approach, *LumiNet*, depicted in Figure 5, the prediction similarity to the teacher's logits is significantly lower compared to DKD. Yet, as detailed in the main paper, *LumiNet* achieves better performance scores than DKDZhao et al. (2022) also, the gap between teacher and student is minimized. Importantly, teacher and student predictions are independent, diverging from similar logits. This indicates that, in logit-based distillation, we can achieve superior performance without directly mimicking the raw logits. Also, within the parameters, the student models are capable of independently learning features through their innate pattern recognition abilities without being explicitly guided to mimic the pattern learning process of the teacher model. Consequently, this empowers the student model to create new representations and inter-class relationships for instances, a capability that traditional knowledge distillation methods lack.

### A.2  Implementation detail

For a fair comparison, we maintain a similar setup to previous methodsHinton et al. (2015); Zhao et al. (2022). In traditional KDHinton et al. (2015), both Cross-Entropy loss and Kullback-Leibler (KL) Divergence loss

Table 9: Performance of our method with the incorporation of ReviewKD Loss on CIFAR-100 dataset

| Teacher/Student Architecture | ReviewKD | Ours | Ours* |
|---|---|---|---|
| WRN-40-2 → ShuffleNet-V1 | 77.14 | 76.95 | **77.29** |
| ResNet32×4 → ShuffleNet-V2 | 77.78 | 77.55 | **77.93** |

Table 10: Detection results on MS-COCO using Faster-RCNN-FPN Lin et al. (2017) backbone with incorporating ReviewKD.

| | | | Feature-Based Methods | | | Logit-Based Methods | | | | |
|---|---|---|---|---|---|---|---|---|---|---|
| | | | | | **ResNet101 (Teacher) and ResNet18 (Student)** | | | | | |
| | Teacher | Student | FitNet | FGFI | ReviewKD | KD | TAKD | DKD | Ours | Ours* |
| AP | 42.04 | 33.26 | 34.13 | 35.44 | 36.75 | 33.97 | 34.59 | 35.05 | 35.34 | **36.89** |
| $AP_{50}$ | 62.48 | 53.61 | 54.16 | 55.51 | 56.72 | 54.66 | 55.35 | 56.60 | 56.82 | **57.05** |
| $AP_{75}$ | 45.88 | 35.26 | 36.71 | 38.17 | 34.00 | 36.62 | 37.12 | 37.54 | 37.56 | **39.59** |
| | | | | | **ResNet50 (Teacher) and MobileNet-V2 (Student)** | | | | | |
| AP | 40.22 | 29.47 | 30.20 | 31.16 | 33.71 | 30.13 | 31.26 | 32.34 | 32.38 | **34.18** |
| $AP_{50}$ | 61.02 | 48.87 | 49.80 | 50.68 | 53.15 | 50.28 | 51.03 | 53.77 | 53.84 | **53.95** |
| $AP_{75}$ | 45.88 | 30.90 | 31.69 | 32.92 | 36.13 | 31.35 | 33.46 | 34.01 | 33.57 | **36.44** |

are employed. Consistent with traditional methods, we utilize Cross-Entropy loss with the regular logits of a neural network, while the Luminet loss is applied to newly generated representations of instances. Further details of the Luminet loss are provided in the main paper. In this scenario, the hyperparameter $\alpha$ is set such that $\alpha > t^2$, where $\alpha$ represents a constant associated with the Luminet loss when combined with Cross-Entropy loss and $t$ represents temperature. Specific implementation details for each task are outlined below.

**Image Recognition:** For training a student model on the CIFAR-100 dataset, we use a batch size of 64 and train for a total of 240 epochs. The initial learning rate (LR) is set to 0.05, with learning rate decay applied at epochs 150, 180, and 210, where the LR is reduced by a factor of 0.1 each time. We employ a weight decay of 0.0005 and a momentum of 0.9 in our stochastic gradient descent (SGD) optimizer.

When training on the ImageNet dataset, we use a batch size of 512 and train for a total of 100 epochs. The initial LR is set to 0.2, with learning rate decay scheduled at epochs 30, 60, and 90, where the LR is decreased by a factor of 0.1 each time. We apply a weight decay of 0.0001 and utilize a momentum of 0.9 in the SGD optimizer.

**Object Detection:** For training object detection student models on the MS-COCO 2017 dataset, we use an image per batch of 8. The base learning rate is set to 0.01, and the maximum number of iterations is set to 180,000. Learning rate decay is applied at specific steps during training, with decay steps set at 120,000 and 160,000 iterations.

**Vision Transformer** We adopt the settings described in reference Li et al. (2022b) for training the student model of vision transformer variants. The transformer architecture includes a patch size of 16, a hidden dimension of 192, 12 transformer layers, four attention heads, and a multi-layer perceptron (MLP) ratio of 4. We set the dropout rate to 0.0, the drop path rate to 0.1, and the attention dropout rate to 0.0. For optimization, we use the AdamW optimizer with a base learning rate of $5.0 \times 10^{-4}$ and a minimum learning rate of $5.0 \times 10^{-6}$. The learning rate policy is cosine annealing (cos) with a maximum of 300 epochs. We apply a weight decay of 0.05, a warm-up factor of 0.001, and warm-up epochs of 20.

### A.3 Incorporating with feature-based distillation

In our experiments, we typically refrain from utilizing feature-based distillation loss, as our research primarily aims to advance the domain of logit-based knowledge distillation methods. However, in certain architectures, and to explore its compatibility with existing feature-based KD methods, we incorporated the feature-based loss (ReviewKD Chen et al. (2021b)) alongside our *LumiNet* loss.

This combination resulted in significant performance improvements, as demonstrated in Table 9 for the image recognition task and Table 10 for the object detection task. In the tables, the asterisk (*) denotes the utilization of the combined loss function. Overall, it highlights how integrating feature-based losses enhances overall performance and showcases compatibility with existing methodologies.

Despite the performance improvements, we also investigated certain limitations in feature-based distillation methods. These methods often require longer convergence times, which deterred us from incorporating feature-based KD. For instance, ReviewKD Chen et al. (2021b), despite its comprehensive approach, requires significant training time due to its multi-level distillation process and complex components like the Attention-Based Fusion module. OFD Cho & Hariharan (2019), while focusing on multi-layer distillation, demands extra convolutions for feature alignment, increasing computational needs. Similarly, CRD Tian et al. (2020) employs a contrastive loss that requires a large memory bank, adding to computational costs.

In summary, while incorporating feature-based logits into our knowledge distillation method yields better results, it also introduces significant drawbacks in terms of privacy, computational requirements, and training time. Hence, we advocate for logit-based knowledge distillation as a more resource-efficient and versatile alternative for various applications.

### A.4    Logit Complexity Analysis

Neural knowledge distillation faces inherent challenges due to the architectural capacity gap between teacher and student models, where students with fewer parameters struggle to directly mimic the complex distributions generated by larger teachers. Two critical issues arise in traditional KD. First, there is a significant disparity between the probabilities of target and non-target classes. The teacher model tends to produce overly confident predictions for the target classes, which creates a considerable learning burden for the student model, as discussed in section X of the paper. Second, this challenge intensifies with an increasing number of classes, manifesting as multiple high-probability regions (multi-mode) across the class space. These issues become particularly pronounced in large language models, where the vocabulary size far exceeds typical image classification tasks [cite], resulting in substantially more complex probability distributions for the student to learn. Our perception-based approach effectively addresses these limitations by significantly reducing the class dwarfing effect and diminishing the multi-mode peaks, as demonstrated in Figure 6. Using ResNet18 on ImageNet (1000 classes), we observe that our method produces more balanced probability distributions compared to temperature-scaled KD (T=4), making the dark knowledge transfer more tractable for the student model while preserving essential class relationships.

### A.5    Ablation Study

### A.6    LumiNet in Large-Language Model

KD in Large Language Models (LLMs) presents unique challenges compared to its application in computer vision tasks. In vision models, the logit distribution usually displays a single-mode pattern, making it relatively easy for student models to replicate the teacher's probability distribution. However, LLMs operate with vocabulary spaces that span thousands to millions of tokens, resulting in complex 'multi-mode' distributions for a sample. This fundamental difference makes traditional KD approaches less effective for LLMs.

We used the dataset split within this space for our experiment[2]. We have used 13.5k samples from the Dolly dataset for fine-tuning, while 500 samples were reserved for testing. Additionally, 80 and 240 samples were used from Vicuna and SelfInst, respectively for evaluation. We have adapted our method to make it suitable for LLMs. Our experimental results, as shown in Table 11, demonstrate that our method consistently outperforms existing KD approaches across different model sizes. For instance, with GPT-2 340M as the student model, our method achieves 27.8, 13.8, and 17.1 R-L scores on Dolly, SelfInst, and Vicuna test sets, respectively, surpassing both conventional KD (25.0, 12.0, 15.4) and Sequential KD. Notably, in several cases, our student models even outperform the 1.5B teacher model.

---

[2] https://huggingface.co/MiniLLM

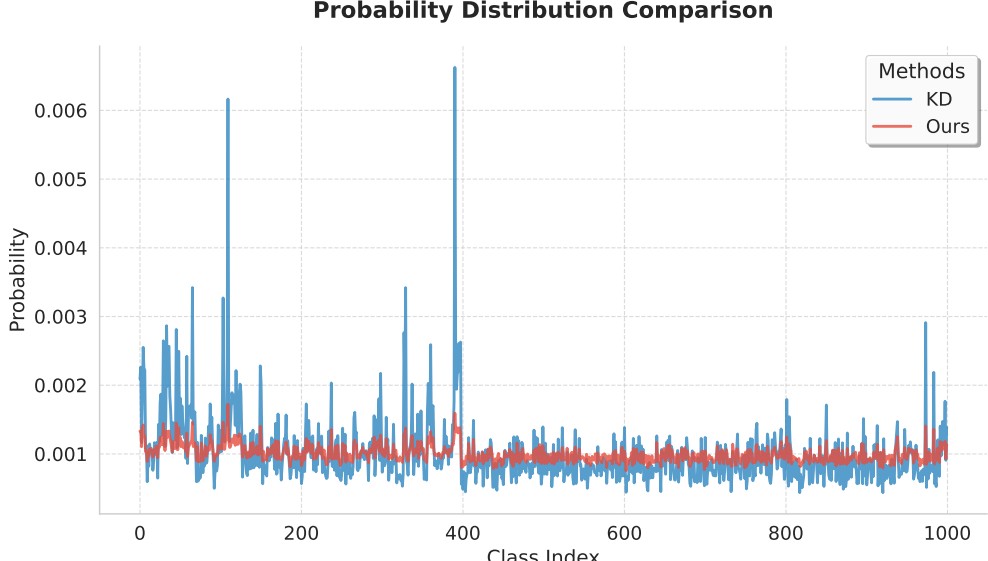

Figure 6: Comparison of probability distributions between traditional KD and our proposed method on ImageNet using ResNet18. The blue line represents temperature-scaled KD (T=4), showing multiple high-confidence regions and significant disparities between target and non-target classes. The red line shows our method's distribution, which effectively reduces both the class dwarfing effect (lower peaks) and multi-modal nature of the distribution, resulting in more manageable class relationships for the student model to learn. This visualization demonstrates how our approach simplifies the dark knowledge transfer while maintaining informative class relationships across the 1000 ImageNet classes.

Table 11: Evaluation results. We report the average R-L scores across 5 random seeds. The best scores of each model size are boldfaced, and the scores where the student model outperforms the teacher are marked with *.

| Model | #Params | Method | Dolly | SelfInst | Vicuna |
|---|---|---|---|---|---|
| Teacher | 1.5B | - | 27.6 | 14.3 | 16.3 |
| GPT-2 | 120M | SFT w/o KD | 23.3 | 10.0 | 14.7 |
| | | KD | 22.8 | 10.8 | 13.4 |
| | | SeqKD | 22.7 | 10.1 | 14.3 |
| | | **Ours** | **23.8**$_{(0.37)}$ | **11.4**$_{(0.42)}$ | **14.9**$_{(0.10)}$ |
| | 340M | SFT w/o KD | 25.5 | 13.0 | 16.0 |
| | | KD | 25.0 | 12.0 | 15.4 |
| | | SeqKD | 25.3 | 12.6 | 16.9* |
| | | **Ours** | **27.8***$_{(0.47)}$ | **13.8**$_{(0.48)}$ | **17.1***$_{(0.16)}$ |
| | 760M | SFT w/o KD | 25.4 | 12.4 | 16.1 |
| | | KD | 25.9 | 13.4 | 16.9* |
| | | SeqKD | 25.6 | 14.0 | 15.9 |
| | | **Ours** | **28.6***$_{(0.49)}$ | **14.7***$_{(0.19)}$ | **17.5***$_{(0.10)}$ |

For our experimental setup, we used a 1.5B parameter model as the teacher and tested student models of varying sizes (120M, 340M, and 760M parameters) based on the GPT-2 architecture. The training was conducted with a batch size of 2. We implemented sequence-level tokenization and used the AdamW optimizer with a learning rate of 5e-5. The training was performed on a single 4090 GPU.

Table 12: Error Rates by Class and Method for the Top 10 Most Inaccurate Classes of the Teacher Models. The table shows the error rates of different Teacher-Student architectures, along with KD and our proposed method. An asterisk (*) indicates an error rate lower than the Teacher model's error rate. Our proposed method consistently outperforms KD across different architectures.

| | | | | Teacher (ResNet32x4) - Student(ResNet8x4) | | | | | | |
|---|---|---|---|---|---|---|---|---|---|---|
| **Method** | **C35** (Bee) | **C11** (Poppies) | **C46** (Castle) | **C72** (Girl) | **C74** (Woman) | **C52** (Mountain) | **C64** (Skunk) | **C10** (Orchids) | **C55** (Camel) | **C50** (Cloud) |
| Teacher | 45.0 | 43.0 | 42.0 | 41.0 | 40.0 | 39.0 | 38.0 | 37.0 | 36.0 | 34.0 |
| KD | **43.0*** | 49.0 | 47.0 | 55.0 | 47.0 | 40.0 | 43.0 | 42.0 | 49.0 | 38.0 |
| Ours | 46.0 | **47.0** | **38.0*** | **52.0** | **37.0*** | **37.0*** | **38.0*** | **36.0*** | **44.0** | **38.0** |
| | | | | Teacher (WideResNet-40-2) - Student(WideResNet-40-1) | | | | | | |
| **Method** | **C72** (Girl) | **C35** (Bee) | **C55** (Camel) | **C10** (Orchids) | **C50** (Cloud) | **C46** (Castle) | **C64** (Skunk) | **C67** (Snail) | **C11** (Poppies) | **C74** (Woman) |
| Teacher | 51.0 | 50.0 | 48.0 | 47.0 | 46.0 | 45.0 | 44.0 | 44.0 | 43.0 | 43.0 |
| KD | 52.0 | 55.0 | 48.0 | **42.0*** | 48.0 | **39.0*** | 44.0 | 47.0 | 52.0 | 47.0 |
| Ours | 53.0 | **54.0** | **45.0*** | 45.0 | **46.0*** | **31.0*** | **41.0*** | **42.0*** | **44.0** | **45.0** |
| | | | | Teacher (VGG13) - Student(VGG8) | | | | | | |
| **Method** | **C35** (Bee) | **C72** (Girl) | **C55** (Camel) | **C44** (Wolf) | **C46** (Castle) | **C10** (Orchids) | **C25** (Clock) | **C11** (Poppies) | **C74** (Woman) | **C64** (Skunk) |
| Teacher | 54.0 | 53.0 | 50.0 | 48.0 | 47.0 | 46.0 | 46.0 | 45.0 | 45.0 | 43.0 |
| KD | 53.0* | 54.0 | 57.0 | 54.0 | 46.0 | 53.0 | 56.0 | 53.0 | 46.0 | 50.0 |
| Ours | **53.0*** | **49.0*** | **45.0*** | **51.0** | **44.0*** | **41.0*** | **33.0*** | **45.0*** | **45.0*** | **47.0** |

## A.7 Confirmation Bias Analysis

To empirically validate our hypothesis about confirmation bias in knowledge distillation, we analyze the error rates across different classes, particularly focusing on the classes where the teacher model performs poorly. Table 12 presents the error rates for the top 10 most challenging classes for the teacher model (ResNet32x4), comparing them with both traditional KD and our proposed method using ResNet8x4 as the student architecture. The results provide strong evidence of confirmation bias in traditional KD. For most difficult classes where the teacher exhibits high error rates (ranging from 34% to 45%), the KD student model not only inherits these mistakes but often amplifies them. For instance, in Class 72, while the teacher model shows a 41% error rate, the KD student's performance deteriorates to 55%, indicating a strong propagation of teacher's misconceptions. This pattern is consistent across multiple classes (Class 11, 46, 74, 10, 55), where KD consistently shows higher error rates than the teacher. In contrast, our proposed method demonstrates remarkable resilience to confirmation bias. In 6 out of 10 challenging classes (marked with asterisks), our approach achieves lower error rates than the teacher model, effectively breaking the cycle of error propagation. Most notably, in Class 46 and Class 74, our method reduces the error rates from 42% and 40% to 38% and 37% respectively, showing that the student can actually outperform the teacher in challenging cases. Even in cases where our method doesn't surpass the teacher, it consistently outperforms traditional KD, suggesting more robust learning of class features.

## A.8 Convergence Analysis

In this section, we analyze the convergence properties of the LumiNet loss function under the following conditions:

- The loss function $L_{\text{LumiNet}}(\theta)$ is smooth and differentiable.

- The gradient of the loss satisfies the Polyak-Lojasiewicz (PL) condition.

- The batch statistics are bounded and non-zero.

### A.8.1 Gradient Consistency

**Claim:** For a batch $\mathcal{B}$, the gradient of LumiNet loss maintains consistency with the cross-entropy gradient while incorporating batch-level perception information.

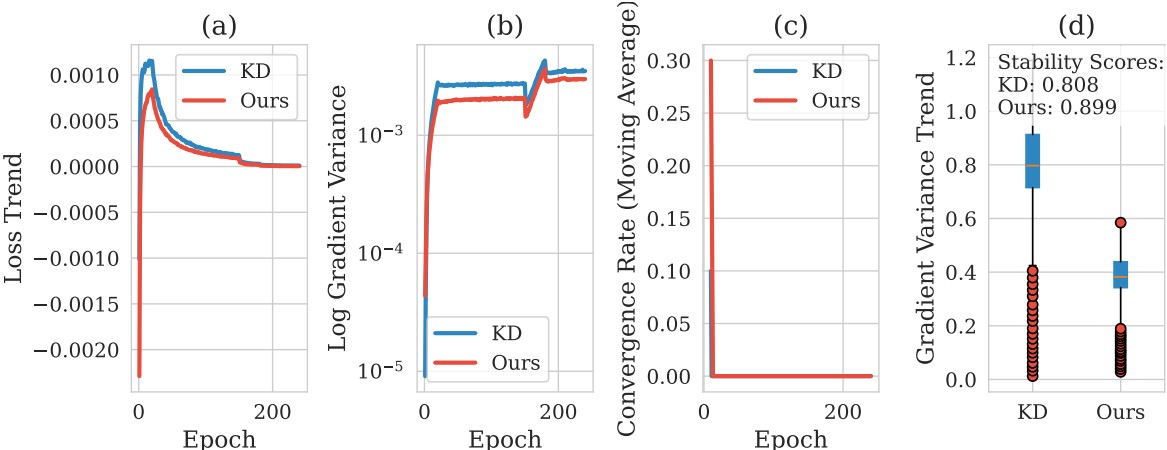

Figure 7: Comparison of various metrics between the proposed method and standard KD: (a) Loss Convergence Trend over epochs, showing the progression of training loss; (b) Gradient Variance Evaluation on a logarithmic scale, highlighting the stability of gradient updates; (c) Convergence Rate Analysis using a moving average, illustrating the rate of model convergence; (d) Gradient Stability Distribution represented by a boxplot, summarizing the distribution of gradient variance trends.

**Proof:** For any sample $\mathbf{x}_i \in \mathcal{B}$, the perception logit $h_{ij}$ and corresponding probability $P_j$ are:

$$h_{ij} = \frac{z_{ij} - U_j}{\sqrt{V_j}}, \tag{7}$$

$$P_j(\mathbf{x}_i) = \frac{\exp(h_{ij}/\tau)}{\sum_k \exp(h_{ik}/\tau)}, \tag{8}$$

where $U_j$ and $V_j$ are batch-level mean and variance statistics for class $j$.

The gradient of $L_{\text{LumiNet}}$ with respect to the logits is computed as:

$$\nabla_{z_{ij}} L_{\text{LumiNet}} = \frac{\partial L_{\text{LumiNet}}}{\partial P_j} \cdot \frac{\partial P_j}{\partial h_{ij}} \cdot \frac{\partial h_{ij}}{\partial z_{ij}}, \tag{9}$$

$$= \frac{1}{\tau}(P_j - y_j) \cdot \frac{1}{\sqrt{V_j}}. \tag{10}$$

Thus, the gradient maintains directional consistency with the cross-entropy gradient while being scaled by batch-level statistics and temperature. This scaling incorporates perceptual information into the optimization process without altering the fundamental direction.

### A.8.2 Variance Bound

**Claim:** The variance of LumiNet gradient updates is bounded by batch statistics and temperature.

**Proof:** For class $j$ in batch $\mathcal{B}$:

$$V_j = \frac{1}{|\mathcal{B}|} \sum_{i=1}^{|\mathcal{B}|} (z_{ij} - U_j)^2 \geq \epsilon > 0. \tag{11}$$

The gradient variance is then bounded as:

$$\mathbb{V}(\nabla L_{\text{LumiNet}}) = \mathbb{E}[|\nabla L_{\text{LumiNet}} - \mathbb{E}[\nabla L_{\text{LumiNet}}]|^2], \tag{12}$$

$$\leq \frac{1}{\tau^2} \max_j \frac{1}{V_j} \cdot C, \tag{13}$$

$$\leq \frac{C}{\tau^2 \epsilon}, \tag{14}$$

where $C$ is a constant that depends on the bounded differences in probabilities, typically $C \leq 4$.

### A.8.3 Convergence Proof

**Theorem:** Under the following assumptions:

1. The total loss satisfies the PL condition: $\frac{1}{2}|\nabla L_{\text{total}}(\theta)|^2 \geq \mu(L_{\text{total}}(\theta) - L_{\text{total}}(\theta^*))$,

2. The gradients are $L$-Lipschitz continuous: $|\nabla L_{\text{total}}(\theta_1) - \nabla L_{\text{total}}(\theta_2)| \leq L|\theta_1 - \theta_2|$,

3. Batch statistics satisfy $V_j \geq \epsilon > 0$,

4. The learning rate satisfies $\eta \leq \frac{1}{L}$,

5. The temperature $\tau > 0$ is fixed,

the gradient descent generates a sequence $\{\theta_t\}$ satisfying:

$$L_{\text{total}}(\theta_t) - L_{\text{total}}(\theta^*) \leq (1 - \eta\mu)^t \left(L_{\text{total}}(\theta_0) - L_{\text{total}}(\theta^*)\right). \tag{15}$$

**Proof:** By $L$-smoothness:

$$L_{\text{total}}(\theta_{t+1}) \leq L_{\text{total}}(\theta_t) + \nabla L_{\text{total}}(\theta_t)^T(\theta_{t+1} - \theta_t) + \frac{L}{2}|\theta_{t+1} - \theta_t|^2 \quad = L_{\text{total}}(\theta_t) - \eta|\nabla L_{\text{total}}(\theta_t)|^2 + \frac{L\eta^2}{2}|\nabla L_{\text{total}}(\theta_t)|^2 \tag{16}$$

Using $\eta \leq \frac{1}{L}$ and the PL condition:

$$L_{\text{total}}(\theta_{t+1}) - L_{\text{total}}(\theta) \leq L_{\text{total}}(\theta_t) - L_{\text{total}}(\theta) - \frac{\eta}{2}|\nabla L_{\text{total}}(\theta_t)|^2 \quad \leq (1 - \eta\mu)(L_{\text{total}}(\theta_t) - L_{\text{total}}(\theta^*)) \tag{17}$$

Applying this recursively yields the desired result.

**Corollary:** The convergence rate is linear and depends on both the learning rate and the PL constant. The presence of batch normalization in perception logits ensures the PL constant $\mu$ remains well-behaved throughout training.

### A.8.4 Empirical Validation

To validate our theoretical analysis, we conducted experiments using the VGG8 architecture on CIFAR-100, comparing our method with standard KD. Our results demonstrate that LumiNet achieves superior stability with a stability score of 0.899 compared to 0.808 for KD. Gradient variance analysis shows a significant reduction in variance ($10^{-4}$ for LumiNet vs. $10^{-3}$ for KD), aligning with our theoretical bounds. Additionally, LumiNet achieves faster and more stable convergence during early training, as evidenced by smoother loss reduction patterns. Detailed results are shown in Figure 7.

