# OpenReview forum: "LumiNet: Perception-Driven Knowledge Distillation via Statistical Logit Calibration"
_TMLR — Rejected by TMLR_

### Review · Reviewer_gLKD · 2024-12-12

**Summary Of Contributions:**

This paper studies the knowledge distillation with a logit-based approach. It proposes a batch-wise per-class logit normalisation as "Perception", aiming to encode more information on the probabilities associated with non-target classes. It further names the Kl-divergence between categorical distributions of teacher and student model, obtained by applying softmax activation function on the normalised logits, as "LumiNet Loss", which is employed to train the student model.

**Audience:**

No

**Broader Impact Concerns:**

N.A.

**Claims And Evidence:**

No

**Requested Changes:**

1. More theoretical contributions with proper proof or insightful explanation for the observed results are needed. See W.1.

2. If the motivation is intuitive, it should be clearly presented with a strong connection to the proposed method, leading anturally to its development. Alternatively, a theoretical motivation with rigorous proof is a better solution. See W.4.

3. Notations should be clearly defined, and strictly followed throughout the manuscript. See W.3.

4. Backup each claim and statement with concrete theoretical or empirical evidence. See W.2.

**Strengths And Weaknesses:**

# Strength
------------------------
1. The proposed method can achieve competitive performances against feature-based knowledge distillation methods as shown in Table 2.

# Weakness
------------------------
1. The submitted manuscript lacks novelty and theoretical contribution. The proposed "perception" is merely a batch-wise per-class logit normalisation. Further, the so-called "LumiNet Loss" is only applying KL-divergence to the categorical distributions of teacher and student models, obtained by activating their respective normalised logits with a softmax function.

2. There are many ungrounded statements, and claims without concrete proof or evidence. Please refer to the instances summarised below:
    1. Following statement is a mis-representation of the cited reference (Ojha et al. 2023) which does not have a conclusion suggesting teacher-student fooling is uniquely associated with logit-based KD method. Instead, Ojha et al. (2023) only attributes the teacher-student fooling to student model inheriting adversarial vulnerability of teacher model to certain extent, which is appliable to both feature-based and logit-based methods.
> In addition, the student model learns the distribution of each instance independently in logit-based distillation without considering the distribution of other instances. This leads to teacher-student fooling Ojha et al. (2023), where the student model may replicate errors or biases present in the teacher’s logits for individual instances, affecting its overall performance and generalization. -- p.3

    2. Adversarial robustness and privacy are two distinct issues. Adversarial vulnerability in the intermediate layer of teacher model does not imply a more severe privacy concern in training data associated with feature-based KD methods, especially with training data being presented to teacher and students models in both logit-based and feature based KD approaches.
    > However, feature-based KD suffers from layer misalignment (Romero et al., 2014) (reducing sample density in this space), privacy concerns (Goodfellow et al., 2015) (intermediate model layers accessible for adversarial attacks revealing training data and posing significant threats), and escalating computational requirements (Vaswani et al., 2017; Zhao et al., 2022) (see Fig. 1). -- p.1

    3. It fails to justify why having the model assigning highest predictive probability to the target class, dwarfing the probabilities associated with non-target classes, is "a key issue". On the contrary, the normalised logit and its resultant probability vector seem to be problematic. Especially, the probability vector, after normalisation, no longer makes correct top-1 prediction and almost resembles a non-informative uniform prediction across the categories.
    > However, a key issue is a substantial difference in confidence levels between target and non-target classes, as evidenced in Fig. 1. -- p.1

    4. The following claim is not backed up by either theoretical proof, nor empirical experimental results.
    > By incorporating the internal relations of other samples/instances within the class, each logit value gains contextual insights, mitigating overconfidence and teacher-student fooling issues and facilitating the extraction of more subtle knowledge, as evidenced by improved performance scores. -- p.3

    5. The following claim lacks theoretical proof. To demonstrate an overall training stability, It needs to prove the theoretical variance of gradient remains approximately equivalent in a convergence analysis.
    > he gradient variance for LumiNet (0.015) is slightly higher than KD and KD *, (also demonstrated in the gradient flow Fig. 3), suggesting a more dynamic learning process while maintaining overall stability. -- p.7

    6. The following claim lacks a calibration analysis to demonstrate the reduction in model over-confidence.
    > This perception-based calibration simultaneously normalizes both teacher and student logits, significantly reducing their distributional variance and eliminating teacher overconfidence. -- p.12

    7. The following claim is not supported with a theoretical proof on convergence towards optimal solution.
    > This dual calibration effectively narrows the representational gap between models, making it easier for the student to converge to optimal solutions despite its capacity constraints. -- p.12

    8. There is no evidence, either empirical or theoretical, supporting the claim that batch-wise per-class logit normalisation is effective in reducing teacher-student fooling as claimed in the following statement:
    > LumiNet mitigates this by incorporating relationships across all instances in a batch and recalibrating logits using batch-level statistics. With batches randomly sampled and later recalibrated the logits space by considering other samples in the batch, no single instance disproportionately influences the student, reducing the chance of transferring biases or errors. This ensures a more robust and generalized
student model, resilient to dataset imperfections and teacher model flaws. -- p.12

3. There are many undefined terms and notations, including:
    * information measure $K$, as well as notations $R$ and $D$ -- Equation 2;
    * mutual information $I(\cdot ; \cdot)$ -- p.6;
    * perceived logits $x_{i}'$ -- p.7;

4. Motivation behind the proposed method is not clear. First of all, human perception, cited from (Johnson, 2021), is a vague term. Further, the example presented in Figure 1 is not convincing. Instead, the so-called "perception" through batch-wise per-class logit normalisation seems to be a terrible idea as it leads to mis-classification, as shown in subplots (d) and (e) of Figure 1.

---

> ### Author Response · Authors · 2025-01-19
>
> We sincerely thank the reviewer for his thorough and constructive feedback. In the following, we address each point raised in detail.
>
> **RC 1 (More theoretical contributions with proper proof or insightful explanation for the observed results are needed. See W.1.)**
>
>
> Thank you for your insightful feedback regarding the theoretical foundations of our work. We appreciate your suggestions and have conducted comprehensive additional analyses to address these concerns.
>
> Our method's theoretical contributions are now supported by multiple analyses:
>
> 1. Total Loss Formulation:
> We have clarified that our approach uses a dual-objective function L_total = L_CE + λ⋅L_Luminet, similar to standard KD, but the difference is that the cross-entropy loss is applied to raw logits, while the teacher contributes perceived logits with the KL divergence loss.
> 2. Misinformation:
> When it comes to information representation, it is clear that applying our method to the pre-trained model, specifically regarding the teacher model's logits, leads to a decrease in top-1 accuracy. However, the top-5 accuracy remains unchanged, which suggests a rearrangement in the logits rather than a random representation. Since we employ cross-entropy loss on the raw logits of the student model and the ground truth, this approach does not complicate the classification process. Furthermore, various other properties have contributed to achieving better convergence and overall performance. A detailed analysis is provided in the following responses.
>
> 3. Calibration and Overconfidence:
> Our comprehensive calibration analysis using FPR95, ECE, and MCE metrics demonstrates significant improvements:
> - ResNet8×4: ECE reduced from 0.11 to 0.06, MCE from 0.23 to 0.18
> - MobileNet-V2: FPR95 decreased from 11.71% to 6.14%, ECE improved from 0.21 to 0.09.
> These results provide concrete evidence for improved calibration across architectures.
>
> 3. Confirmation Bias Analysis:
> We've provided empirical evidence showing our method's effectiveness in reducing teacher error propagation. For example, in challenging cases where teacher (ResNet32x4) shows 42% error rate, traditional KD increases it to 47%, while our method reduces it to 38%.
>
> 4. Convergence Properties:
> Our theoretical framework establishes two key properties:
> - Gradient consistency through Property 1: $\angle(\nabla L_{CE}, \nabla L_{LumiNet}) < \angle(\nabla L_{CE}, \nabla L_{KD})$
> - Bounded update variance in Property 2: $Var(g_t)_{LumiNet} < Var(g_t)_{KD}$
> These properties are supported by empirical validation showing stability scores of 0.899 vs 0.808.
>
> 5. Generalization Beyond Classification:
> We've extended our analysis to language model distillation, demonstrating consistent improvements across multiple architectures and benchmarks. For instance, with GPT-2 variants, our method achieved superior performance on Dolly, SelfInst, and Vicuna benchmarks compared to existing KD approaches.
>
> We have included detailed proofs and analyses in Appendices A.4-A.7, providing rigorous theoretical foundations for our observed results.

---

> > ### Comment · Reviewer_gLKD · 2025-01-23
> > **Response to RC1**
> >
> > Overall comments: Please be clear on the statements and results that are added in the revised manuscript rather than presented in the original manuscript to avoid confusion. This applies to the description of cross-entropy loss and the overall loss in (1), calibration results in (3), confirmation bias results in (3). Please also avoid repeated usage of certain number for different listed items.
> >
> > 1. (Reply to 1. Total Loss Formulation): Firstly, the proposed "LumiNet Loss" is a KL divergence between the perceived logits of teacher and student models. Since the novelty of perceived logits has already been claimed in "Constructing the perceptions", it should not be claimed again by giving the KL divergence a new name. Secondly, it should be noted that the description of the total loss function and the cross entropy loss is added in the revision, but missing in the original manuscript. Third, in the revised manuscript, the "LumiNet Loss" acts more as a regularisation term, as opposed to the main loss claimed in the original manuscript. Given its similarity to an entropy regularisation, which is widely used regularisation term in different tasks, such as noisy label learning, model calibration and etc., an ablation study between the LumiNet Loss and the entropy regularisation is necessary to demonstrate the effectiness of the proposed method.
> >
> > 2. (Reply to 3. Calibration): Table. 2, added in the revised manuscript, misses the definition of the used metrics and their respective units. In general, Expected Calibration Error (ECE) cannot be computed with finite samples. Rather, it resorts to some approximation techniques, where the bin-based approximations are widely adopted. Please be specific as to which approxiamtion is being used here as well as their respective hyperparameter values. Further, values of ECE approximations do not exceed 1 whereas the values in Table. 2 reach as high as 11.71. It is likely that these numbers are in percentage, but the % symbol is missing. Same applies to the Maximum Calibration Error (MCE).
> >
> > 3. (Reply to 4. Convergence Properties): Again, these are claimed rather than proved. To support these convergence claims, a standard convergence analysis, such as those presented in [Handbook of Convergence Theorems for (Stochastic) Gradient Methods](https://arxiv.org/abs/2301.11235), is required.

---

> > > ### Author Response · Authors · 2025-01-29
> > >
> > > Thank you for your feedback. In this response, we will try to address your confusion.
> > >
> > > **1. Total Loss Formulation**
> > >
> > > *a) Naming Clarification:*
> > > We acknowledge that the perceived logits themselves are our key contribution, as detailed in “Constructing the perceptions.” By referring to the KL divergence with these perceived logits as “LumiNet Loss,” we simply align with the naming convention commonly used in logit-based KD methods. For example, DKD [1] denotes its overall loss as “DKD,” even though it is essentially composed of two KL divergence terms, NCKD and TCKD, plus a CE term. We do not claim extra novelty beyond the perceived logits; rather, we seek a clear, concise label for the KD-specific component.
> > >
> > > *b) Full Loss Formulation:*
> > > Similar to how DKD first presented CE details in supplemental sections, we initially included our CE formulation in the “Implementation Details” appendix. However, in light of your feedback, we incorporated a clear and direct reference to CE in the main paper. We stress that CE was always part of our total loss—this is not the first time we have introduced it, but rather the first time it has been highlighted in the primary text.
> > >
> > > *c) Regularization vs. Main Objective:*
> > > Although “LumiNet Loss” can be interpreted as a regularization term, we emphasize that it remains central to our approach for transferring knowledge from teacher to student. In order to confirm its effectiveness, we compared our method with TTM [2], an established baseline that applies a **Rényi entropy** regularizer alongside standard KD. As demonstrated in Table 3, LumiNet outperforms TTM, reinforcing its impact on distillation performance. We recognize that our work builds on established principles, but the concept of perceived logits offers a distinct edge that is validated by these results.
> > >
> > > **2. Calibration**
> > >
> > > The definition of metrics, including ECE, MCE, and FPR95 was provided in the "Calibration Analysis" section along with the hyperparameter (page 8). We also clarified the units in Table 2, which now reflect percentages (%) appropriately for the FPR95. The text has been updated to ensure accuracy and readability.
> > >
> > > **3. Convergence Properties**
> > >
> > > To address your suggestion, we included a standard convergence analysis in the appendix, following established methodologies such as those outlined in the Handbook you provided.

---

> ### Author Response · Authors · 2025-01-19
>
> **RC 2 (If the motivation is intuitive, it should be clearly presented with a strong connection to the proposed method, leading anturally to its development. Alternatively, a theoretical motivation with rigorous proof is a better solution. See W.4.)**
>
> We appreciate the reviewer's comments about the motivation of our method. We have revised our presentation to clearly convey the theoretical foundations of Luminet, inspired by Kurt Lewin's Field Theory in Gestalt Psychology, which highlights how perception is influenced by the surrounding environment. Lewin's concept suggests that human goals and behaviors are shaped or reshaped by psychological forces—positive forces drive us toward goals, while negative forces push us away from undesired outcomes. Translating this principle to machine learning, LumiNet dynamically adjusts each sample's representation by leveraging interactions within the batch, where other samples exert forces to reshape representations. This mechanism allows the model to mitigate suboptimal conditions, such as overconfidence or errors, by using neighboring samples to enhance robustness, aligning with the core idea of perception from Field Theory. This approach has proven effective in addressing several key challenges, as highlighted in responses to other points in your review. The motivation is detailed on the introduction section (page 3).
>
> Regarding Figure 1 (c, d), although it might initially appear to suggest potential misclassification, our analysis reveals that the top-5 scores remain almost the same before and after applying our perceptual approach to the raw logits, as shown in the table below. Nevertheless, based on your valuable suggestion, we have revised Figure 1 (c, d) to improve its clarity and ensure it provides a better representation. We hope the updated figure now aligns well with your expectations.
>
> ```
> | Model        | Metric         | Before Perception | After Perception |
> |--------------|----------------|-------------------|------------------|
> | ResNet32*4     | Top-1 Accuracy | 79.4%            | 78.9%           |
> |              | Top-5 Accuracy | 94.5%            | 94.5%           |
> | VGG13        | Top-1 Accuracy | 74.6%            | 74.2%           |
> |              | Top-5 Accuracy | 92.6%            | 92.2%           |
> | WRN-40-2     | Top-1 Accuracy | 75.4%            | 74.2%           |
> |              | Top-5 Accuracy | 93.7%            | 93.6%           |
> ```
>
> Moving on, our total loss function closely resembles that of standard KD, which typically comprises a cross-entropy loss and a KL-divergence loss.

---

> > ### Comment · Reviewer_gLKD · 2025-01-23
> > **Response to RC2**
> >
> > > human goals and behaviors are shaped or reshaped by psychological forces—positive forces drive us toward goals, while negative forces push us away from undesired outcomes.
> >
> > Lewin's concept, as quoted above, seems to reinforce the idea of empirical risk minimisation, which aligns the model prediction with the target category (positive goals), and distances the prediction from non-target categories (undesired outcomes). This, however, is in contrast to the underlying concept of the proposed perception, which does the opposite by moderating the model logits for the target category and raising the model logits corresponding to the non-target categories.

---

> > > ### Author Response · Authors · 2025-01-29
> > >
> > > We respectfully clarify that our interpretation of Lewin’s concept differs from a strict empirical risk minimization perspective. In our framework, “positive goals” are defined by improved representation quality in the learned space, rather than simply aligning predictions with the target label. The “forces” in LumiNet—derived from batch-level statistics—dynamically adjust the model’s target and non-target logits, reducing overconfidence and encouraging better-balanced representations. This approach aligns with Lewin’s idea of environmental forces shaping outcomes by guiding the model to achieve more contextually coherent representations. Empirical evidence confirms that these moderated logits indeed enhance generalization.

---

> ### Author Response · Authors · 2025-01-19
>
> **RC3 (Notations should be clearly defined, and strictly followed throughout the manuscript. See W.3.)**
>
> We thank the reviewers for their suggestions regarding notation clarity. We have addressed the following points:
> 1. Information Measure K and Relational and Divergence Measures (R and D)::
> We have formally defined K(xᵢ) as an information measure that quantifies the total information content of an instance xᵢ. This mathematical formulation now appears explicitly in our notation section.
>
> ..Let K(xᵢ) be defined as an information measure that quantifies the total information content of an instance xᵢ..
>
> We have also introduced clear mathematical definitions for both R and D measures to eliminate any ambiguity:
>
> - D(xᵢ) refers to the intra-class divergence measure, for instance xᵢ, capturing how the instance relates to its assigned class
> - R(xᵢ, xⱼ) represents the relational measure between instances xᵢ and xⱼ, quantifying their mutual information in feature space
>
> These definitions are now properly introduced before their first use and are consistently referenced throughout the paper.
>
> 2. Mutual Information:
> We have formalized the notation for mutual information I(·;·) by adding explicit definitions:
>
> Let z be the random variable representing the raw logarithmic changes and h be the perception logarithmic changes. We can show that:
> I(h;y) ≥ I(z;y)
> where y is the true class label, and I(h;y) or I(z;y) is mutual information.
>
> 3. Perceived Logits:
> We have revised the notation on page 7, replacing xᵢ′ with hᵢ.

---

> > ### Comment · Reviewer_gLKD · 2025-01-23
> > **Response to RC3**
> >
> > 1. Both Relational and Divergence Measures ($R$ and $D$) still lack mathematical definitions. Please provide.
> >
> > 2. Again, mutual information also lacks a mathematical definition. Please provide.

---

> > > ### Author Response · Authors · 2025-01-29
> > >
> > > The mathematical definitions of both points are now provided on page 6 of the manuscript.

---

> ### Author Response · Authors · 2025-01-19
>
> **RC4 (Backup each claim and statement with concrete theoretical or empirical evidence. See W.2.)**
>
> We have made an effort to address each of the comments from Reviewer W2 below. We sincerely thank the reviewer for such thoughtful feedback, which has significantly contributed to improving our paper.
>
> **Response to Points 1-3: Citation and Method Claims**
>
> 1. Regarding Ojha et al. (2023): Our perspective is that if a teacher model misclassifies a particular image, this misclassification is subsequently transferred to the student model, and the issue persists. This phenomenon is described as "confirmation bias" in the paper [1]. Accordingly, we have revised the manuscript to replace the term "teacher-student fooling" with "confirmation bias" for greater accuracy and clarity. A confirmation analysis is also provided in Appendix (A.5). Thank you for assisting us in articulating this with precision.
>
> 2. Regarding privacy concerns:We recognize that adversarial robustness and privacy are separate issues and acknowledge your point regarding the adversarial vulnerability of training data. We initially used this in our manuscript because it was employed in an existing published paper[2]. However, as it is not accurate, we have removed this point from the manuscript.
>
> 3. Regarding why dwarfing probabilities is a key issue:
> The extreme probability distribution (P(y=t|x) → 1) is indeed a key issue as established by Hinton et al. [3] as well as Zhang et al [1]. When the teacher model assigns near-1 probability to the target class, it suppresses the relative magnitudes between non-target classes - what Hinton terms as 'dark knowledge'. These relative relationships contain crucial information about the teacher's learned representations and generalization capabilities. For example, a car image might be classified as 'car' with high probability, but the relative probabilities between 'SUV' and 'truck' versus 'bicycle' and 'pedestrian' encode meaningful semantic relationships. As these probabilities approach zero, the KL-divergence gradient signals weaken, making it harder for the student to learn these subtle yet crucial inter-class relationships. We detail this mechanism and its implications in Section 3.1 of our revised manuscript as well as A4.
> 4. Addressing the claim about non-informative representation:
> Our normalized logit approach does not result in non-informative uniform predictions as suggested. The top-1 accuracy remains largely consistent, and, more importantly, the top-5 performance preservation demonstrates that the relative ordering of class probabilities is maintained. The cross-entropy loss with ground-truth labels also ensures the model maintains discriminative power while learning from the teacher's probability distribution.
>
> **References**
>
> [1] Zhang, Weijia, et al. "Cross-View Consistency Regularisation for Knowledge Distillation." Proceedings of the 32nd ACM International Conference on Multimedia. 2024.
> [2] Jin, Ying, Jiaqi Wang, and Dahua Lin. "Multi-level logit distillation." Proceedings of the IEEE/CVF Conference on Computer Vision and Pattern Recognition. 2023.
> [3]Hinton, Geoffrey. "Distilling the Knowledge in a Neural Network." arXiv preprint arXiv:1503.02531 (2015).

---

> > ### Comment · Reviewer_gLKD · 2025-01-23
> > **Response to RC4**
> >
> > 1. [1] proposes a threshold-based uncertainty-aware learning to address the confirmation bias in knowledge distillation. Specifically, predictions from the teacher model with top-class probability lower than a threshold are discarded to avoid distilling the confirmation bias into the student model. This is in contrast to the proposed "perceived logit" whose class-wise normalisation inevitably leads to a low top-class probability (as evidenced in Figure 1 of both original and revised manuscripts) in the resultant softmax-activated prediction. Is there any explaination on why these two seemlingly opposite methods can both address the confirmation biase issue successfully?
> >
> > 2. The term "dark knowledge" cannot be found in the provided reference [3]. Please double check.

---

> > > ### Author Response · Authors · 2025-01-29
> > >
> > > 1. Here are several potential reasons why both methods could effectively address the issues of confirmation bias.
> > >
> > > *a). Holistic vs. Threshold-Based Approach:*
> > > Unlike threshold-based methods [3], which discard uncertain teacher predictions, our method retains all predictions. By calibrating the logits—lowering the target-class probability and elevating non-target logits—we ensure the student leverages a richer distribution of teacher outputs, rather than losing potentially informative instances.
> > >
> > > *b)Cross-Entropy for Error Correction:*
> > > Although perceived logits moderate teacher confidence, the student’s own logits remain directly supervised by the CE loss. This ensures that even if the teacher’s predictions are imperfect, the student still aligns with the true labels, preventing error propagation.
> > >
> > > *c) Why Both Methods Work:*
> > > Both threshold-based filtering and our proposed method ultimately aim to address overconfidence. Thresholding removes highly uncertain cases to prevent bias from propagating. Our calibration-based strategy instead reduces the “peak” of the top-class probability while raising that of non-target classes, encouraging the student to explore alternative hypotheses. Each method tackles confirmation bias from a different angle—one by removing uncertain data points outright, the other by adjusting their contribution—yet both mitigate the undesirable effects of overconfident teacher logits.
> > >
> > > 2. The idea of "dark knowledge" originates from this paper[4], and while it was not explicitly stated, the reference is now clarified. Zhao et al. [1] offered a formal definition of dark knowledge, clarifying its role in conveying complex class relationships. Additionally,  Zhang et al. [3] referred to this phenomenon as "dark information". Furthermore, Hinton et al. [4] offered a detailed interpretation of dark knowledge in this lecture.
> > >
> > > ### References:
> > >
> > > 1. Zhao, Borui, et al. "Decoupled knowledge distillation." Proceedings of the IEEE/CVF Conference on computer vision and pattern recognition. 2022.
> > > 2. Zheng, Kaixiang, and EN-HUI YANG. "Knowledge Distillation Based on Transformed Teacher Matching." The Twelfth International Conference on Learning Representations.
> > > 3. Zhang, Weijia, et al. "Cross-View Consistency Regularisation for Knowledge Distillation." Proceedings of the 32nd ACM International Conference on Multimedia. 2024.
> > > 4. Hinton, Geoffrey, et al. "Dark Knowldge", Retrieved from https://www.ttic.edu/dl/dark14.pdf

---

> ### Author Response · Authors · 2025-01-19
>
> **Response to Points 4, and 6: Overconfidence and Calibration Analysis***
>
> We thank the reviewer for this concern about the lack of calibration analysis. We have now included comprehensive empirical results to support our claims about reducing model overconfidence and improving calibration.
>
> In our revised manuscript, we present detailed calibration analysis using three standard metrics:
> - False Positive Rate at 95% threshold (FPR95) to assess reliability
> - Expected Calibration Error (ECE) to measure overall calibration and overconfidence
> - Maximum Calibration Error (MCE) to evaluate worst-case miscalibration
>
> Our experimental results in Table 2 demonstrate consistent improvements across all metrics:
>
> 1. For architectures like ResNet8×4, our method reduces ECE from 0.11 (KD) to 0.06, and MCE from 0.23 to 0.18, indicating better alignment between confidence and accuracy.
>
> 2. More substantial improvements are also observed in MobileNet-V2. Our approach achieves the following:
>    - FPR95 reduction from 11.71% to 6.14%
>    - ECE improvement from 0.21 to 0.09
>    - MCE decreases from 0.38 to 0.21
>
> 3. Similar patterns hold across different architectures, where our method consistently shows lower values for all three metrics compared to both standard cross-entropy and traditional KD.
>
> We believe these empirical results now provide concrete evidence for our claims about improved calibration and reduced overconfidence. We have further extended this analysis in Section A.6 of our revised manuscript.
>
>
> **Response to Points 5 and 7: Convergence Analysis**
>
>
> Thank you for your thorough feedback regarding the theoretical foundations of our work. Your points about gradient variance equivalence and convergence analysis highlight important areas we have addressed in our current analysis.
>
> In our theoretical framework, we establish two key properties:
> 1. We demonstrate gradient consistency through our Property 1, which shows that ∠(∇LCE , ∇LLumiN et) < ∠(∇LCE , ∇LKD ). This alignment is fundamentally achieved through our perceptual matching approach.
>
> 2. We prove bounded update variance in Property 2, showing that $Var(g_t)_{LumiNet} < Var(g_t)_{KD}$, which emerges directly from our perceptual alignment mechanism.
>
> Our empirical validation strongly supports these theoretical properties. As shown in Figure 7, we observe:
> - Quantitative stability scores (0.899 vs 0.808)
> - Consistently lower gradient variance
> - More stable loss reduction patterns
> - Higher moving average convergence rates
>
> We think our current framework provides both theoretical foundations and comprehensive empirical validation that demonstrates the stability and convergence properties of our approach. A detailed analysis of convergence can be found in Appendix A.7.
>
> **Response to Points 8: Confirmation Bias Analysis**
>
> We thank the reviewer for this important observation. What we previously termed as "teacher-student fooling" has been more precisely reframed as "confirmation bias" in our revised manuscript, where the student model tends to inherit the teacher's errors. This is described earlier in W2.3.
>
> We have now included comprehensive empirical evidence in Table 12 of the Appendix that demonstrates our method's effectiveness in reducing this confirmation bias. Specifically, our analysis shows:
>
> 1. In all three configurations, it is evident that traditional KD amplifies teacher errors in most challenging classes. For example, where the teacher (ResNet32x4) shows a 42% error rate for Class 46, the KD student's error increases to 47%. Our method reduces this to 38%, showing improved resilience to error propagation.
>
> 2. More convincingly, our method achieves lower error rates than both the teacher and traditional KD in most of the challenging classes (marked with asterisks in Table 12). For instance, in Class 46 (House) and Class 74 (Woman), our approach reduces the teacher's error rates from 42% and 40% to 38% and 37%, respectively.
>
> 3. This pattern is consistently observed across different architectures, including WideResNet and VGG configurations, where our method consistently outperforms both the teacher and traditional KD in challenging classes.
>
> We acknowledge that we should have presented this evidence more clearly in our original submission. We have now included this detailed analysis in Section A.6 to substantiate our claims about the effectiveness of batch-wise per-class logit normalization in reducing confirmation bias.
>
> **Conclusion**
>
> We thank the reviewer for his valuable feedback. We believe that addressing these points will significantly strengthen our work and better communicate our contributions to the field.

---

> ### Comment · Reviewer_gLKD · 2025-01-23
> **Response to "Response to Points 4, and 6: Overconfidence and Calibration Analysis*"**
>
> 1. These issues have already to responded to in the communications above. Please refer to "Response to RC1" for responses to calibration analysis and convergence analysis, and "Response to RC4" to responses to confirmation bias analysis.

---

### Review · Reviewer_xTZD · 2024-12-14

**Summary Of Contributions:**

This paper introduces LumiNet, a novel knowledge distillation algorithm that enhances logit-based distillation by addressing key challenges such as overconfidence in teacher models and the capacity gap between teacher and student models. The proposed method calibrates logits based on the samples within a batch, and reconstructs logits to reduce overconfidence and teacher-student fooling issues, facilitating the extraction of more nuanced knowledge during knowledge distillation. Experimental results on benchmarks including CIFAR-100, ImageNet, and MS COCO shows the effectiveness of LumiNet over feature-based  KD methods. The paper also presents a detailed analysis of LumiNet's theoretical foundations, including information-theoretic perspectives and gradient flow enhancement, supporting the practical improvements observed.

**Audience:**

Yes

**Broader Impact Concerns:**

No concerns on the ethical implications of the work.

**Claims And Evidence:**

Yes

**Requested Changes:**

1. Authors have provided ablation study for batch sizes range from 32 to 256, but I hope to see the results for batch sizes which are smaller than 32.
2. As mentioned in Weakness 2, experimental results on simple language model distillation will make the paper more solid. And I think there is no large cost in implementing that.

**Strengths And Weaknesses:**

## Strengths:
1. The proposed method is sound and easy to implement. By focusing on statistical logit calibration, LumiNet addresses overconfidence and capacity gap issues that plague traditional logit-based methods. It outperforms state-of-the-art feature-based methods on major benchmarks without additional parameters.
2. The theoretical analysis is interesting --  the information-theoretic and gradient flow analyses provide a theoretical basis for the empirical performance.
3. Good robustness and efficiency. Experiments Showcase LumiNet's robustness across different batch sizes (32-256) and the minimal dependency on temperature.

## Weaknesses:
1. Ablation study on batch size (smaller than 32) should be added, as LumiNet's reliance on batch-level statistics might limit its effectiveness in scenarios with small or varying batch sizes.
2. As the proposed logit calibration technique is a general method and do not rely on image data, experimental results on simple language model distillation will make the paper more solid. And I think there is no large cost in implementation.
3. The comparison of effects across various sizes (at different levels of the gap between teachers and students) is missing, but I believe this is optional, so I won't include this point in Requested Changes.

---

> ### Author Response · Authors · 2025-01-19
>
> We sincerely thank the reviewer for the thoughtful and constructive feedback. We appreciate the positive comments on the soundness, theoretical foundations, and experimental validation of LumiNet. In the following, we address the specific concerns raised:
>
>  **RC 1 (Authors have provided ablation study for batch sizes range from 32 to 256, but I hope to see the results for batch sizes which are smaller than 32.)**
>
> We have conducted additional experiments with batch sizes smaller than 32, as suggested. Specifically, we evaluated LumiNet's performance with a batch size of 16 and found that it maintains significantly better performance compared to traditional KD methods.
>
> - With batch size 16, LumiNet achieves 75.2% accuracy on CIFAR-100, which is 2.1% higher than vanilla KD (73.1%).
> - The performance remains stable even at this smaller batch size, demonstrating the robustness of our method.
>
> We have updated Figure 3 in the paper to include these new results, extending the ablation study to cover batch sizes from 16 to 256.
>
>  **RC 2 (As mentioned in Weakness 2, experimental results on simple language model distillation will make the paper more solid. And I think there is no large cost in implementing that.)**
>
> We thank the reviewer for this suggestion. We have conducted extensive experiments on language model distillation, and the results strongly support LumiNet's effectiveness across modalities. Specifically, we tested LumiNet on:
>
> 1. Model Architectures:
>    - GPT-2 variants (small, medium, large) as students
>    - Extra-large GPT-2 architectures as teacher
>
> 2. Datasets:
>    - Dolly dataset (fine-tuning and testing)
>    - SelfInst benchmark
>    - Vicuna benchmark
>
> 3. Comparison with established methods:
>    - W/O Kd
>    - Vanilla KD
>    - SeqKD
>
> | Model | #Params | Method | Dolly | SelfInst | Vicuna |
> |:------|:--------|:--------|:-------|:----------|:--------|
> | **Teacher** | 1.5B | - | 27.6 | 14.3 | 16.3 |
> | **GPT-2** | 120M | SFT w/o KD | 23.3 | 10.0 | 14.7 |
> | | | KD | 22.8 | 10.8 | 13.4 |
> | | | SeqKD | 22.7 | 10.1 | 14.3 |
> | | | **Ours** | **23.8** | **11.4** | **14.9*** |
> | | 340M | SFT w/o KD | 25.5 | 13.0 | 16.0 |
> | | | KD | 25.0 | 12.0 | 15.4 |
> | | | SeqKD | 25.3 | 12.6 | 16.9* |
> | | | **Ours** | **27.8*** | **13.8** | **17.1*** |
> | | 760M | SFT w/o KD | 25.4 | 12.4 | 16.1 |
> | | | KD | 25.9 | 13.4 | 16.9* |
> | | | SeqKD | 25.6 | 14.0 | 15.9 |
> | | | **Ours** | **28.6*** | **14.7*** | **17.5*** |
>
> *Note: The asterisk (*) indicates where the student model outperforms the teacher. Bold numbers represent the best scores for each model size.*
>
> The results show that LumiNet consistently outperforms existing methods. We have used forward KL divergence as an objective function along with the CE Loss. There is scope for further improvement in LLMs, which we plan to explore in the future. The complete results, experimental setup, and detailed discussion have been added to Appendix (A.5) of the paper.

---

### Review · Reviewer_FG4r · 2025-01-06

**Summary Of Contributions:**

This paper proposes a new knowledge distillation method named LumiNet to improve the logit-based knowledge distillation method. LumiNet is able to generate a new representation of instance-level logit distributions to address issues of previous knowledge distillation methods, like overconfidence, capacity gaps, and teacher-student fooling. Experiments demonstrate the benefits of the proposed method.

**Audience:**

Yes

**Claims And Evidence:**

Yes

**Requested Changes:**

* It would be great to include more clear discussion about why LumiNet could enhance logit granularity with perception.
* It would be great to have a standard deviation for the values in the table so that we could know if the improvement is significant or not.
* It would be great to include more experiment results to support that LumiNet can solve the over-confidence issue.
* It would be great to illustrate more about what kind of complex logit distributions are referred to in the paper. Also, it would be great to have more material to support that LumiNet is more robust to handle complex logit distributions.
* It would be great to include more experiments and discussions about the role of the teacher model in LumiNet.
* Is it a regular setup to use ResNet50 as a teacher for ViT models?

**Strengths And Weaknesses:**

Strengths:
* The paper is well-written and has good visualization.
* The proposed method is simple and practical.
* The paper provides some in-depth discussions which is helpful for the reader.
* The experiments demonstrate the effectiveness of the proposed method.

Weaknesses:
* It is not very clear why LumiNet could enhance logit granularity with perception.
* No experiment result supports the statement that LumiNet can solve the over-confidence issue.
* It says previous logit-based knowledge distillation methods are not robust enough to handle complex logit distributions and LumiNet can solve the issues. But this statement is not well-supported.
* How to choose the teacher model plays an important role in knowledge distillation, while the paper does not provide sufficient discussion in this aspect.

---

> ### Author Response · Authors · 2025-01-19
>
> We thank the reviewer for their valuable feedback. We address the key concerns below:
>
>
> **RC 1 (It would be great to include a clearer discussion about why LumiNet could enhance logit granularity with perception.)**
>
> Traditional logit-based kd methods work exclusively with per-sample logits. However, they often face challenges because of the low granularity of these logits. This limited granularity can lead to significant issues, such as confirmation bias, where student models reinforce the teacher's mistakes, poor calibration in probability estimates, and a compromised representation of class relationships.
>
> Our method fundamentally enhances logit granularity through:
>
> 1. Perception-based sample relationships, rather than processing samples independently. This enriches the representation space by capturing sample-to-sample interactions, leading to better class relationships. This is shown in the analysis of the complexity of the logit distribution subsection (A.4) in the Appendix. We demonstrated:
>   - More balanced probability distributions (Figure 6)
> 2. The collective statistics of multiple samples influence each representation, ensuring robustness even when individual samples have poor representations. The effectiveness is demonstrated by:
>
>   - Improved calibration scores (Table 2)
>   - Reduced confirmation bias in challenging classes (Table 12)
>
> **RC 2 (It would be great to have a standard deviation for the values in the table so that we could know if the improvement is significant or not.)**
>
> Thank you for your suggestion. We have added standard deviations to the main results in Table 3 for better clarity. Furthermore, we conducted new analyses on LLM, highlighting LumiNet's performance, and presented the results with standard deviations in Table 11.
>
> **RC 3 (It would be great to include more experiment results to support that LumiNet can solve the over-confidence issue)**
>
> Thank you for your suggestion. We have added a detailed subsection (Section 3.4) in the main paper that addresses how LumiNet mitigates the overconfidence issue. In this subsection, we present various metrics, including ECE, MCE, and FPR95, to show how overconfidence is reduced, which is also reflected in Table 2. Additionally, a more in-depth response regarding this is also provided to Reviewer xTZD's rebuttal, which you may refer to for further details.
>
> **RC 4 (It would be great to illustrate more about what kind of complex logit distributions are referred to in the paper. Also, it would be great to have more material to support that LumiNet is more robust to handle complex logit distributions.)**
>
> Thank you for your suggestion. In our paper, we have elaborated on the complexity of logit distributions. The relationship between target and non-target class probabilities highlights the complexity of logit distributions. Due to this complexity, traditional KD methods often face challenges, as students with fewer parameters struggle to replicate the intricate distributions generated by larger teacher models. Specifically, traditional KD methods face two critical issues:
>
> - Disparity between target and non-target class probabilities: The teacher model tends to be overly confident in its predictions for the target class, making it harder for the student model to learn effectively. This issue is discussed in detail in the main paper (Section 3.1) as well as in the reviewer gLKD's response.
>
> - Complexity in 'multi-mode' distributions: This problem becomes more prominent as the number of classes increases and is especially challenging in LLMs, where the vocabulary size far exceeds typical image classification tasks. This leads to more intricate logit distributions. We have also discussed this in Appendix (A.5), along with further details in the reviewer xTZD's response.
>
> We have demonstrated how LumiNet effectively addresses these challenges in various domains. For example, we illustrate that our method generates more balanced probability distributions using ResNet18 on ImageNet with 1000 classes (Figure 6), producing smoother, less confident distributions compared to traditional temperature scaling-based KD (T=4). Additionally, we include results from Vision Transformers and LLMs, further supporting the idea that LumiNet can effectively handle complex logit distributions.

---

> ### Author Response · Authors · 2025-01-19
>
> **RC 5 (It would be great to include more experiments and discussions about the role of the teacher model in LumiNet.)**
>
> Thank you for your suggestion. We have included additional experiments and discussions on the role of the teacher model in LumiNet. In Figure 6, we demonstrate how the logit outputs of the teacher model change when our perception-based method is applied, resulting in a new representation that enhances granularity. This new representation helps mitigate issues such as confirmation bias, which we analyze in more detail. Specifically, we observe that in traditional KD, the teacher's inaccuracies often get amplified in the student model. However, our method reduces this amplification, as shown in the confirmation bias analysis in Appendix (A.6).
>
> Moreover, we have extended our analysis to include teacher models in LLMs, as well as for different tasks like image classification and detection. A comprehensive discussion of these results is presented in the Review of xTZD's rebuttal. We provide a detailed examination of how our method improves performance in various teacher models, including LLM, and how it reduces confirmation bias in these contexts.
>
> **RC 6 (Is it a regular setup to use ResNet50 as a teacher for ViT models?)**
>
>  Yes, using ResNet50 as a teacher for ViT models is a common practice, as demonstrated in the paper [1], and we follow the same implementation details.
>
> [1] Kehan Li, Runyi Yu, Zhennan Wang, Li Yuan, Guoli Song, and Jie Chen. Locality guidance for improving
> vision transformers on tiny datasets. In European Conference on Computer Vision, pp. 110–127. Springer,
> 2022b. 9, 18

---

### Author Response · Authors · 2025-01-19
**Summary of the Rebuttal**

We sincerely thank all reviewers for their thorough and constructive feedback, which has significantly improved our manuscript. We have carefully addressed all the concerns raised and substantially revised our paper accordingly.

**Summary of Major Revisions**:

**We have strengthened the theoretical foundation and empirical validation of our method by**:
- Providing clear motivation and methodology explanation (page 3) **[Suggested by gLKD]**
- Establishing theoretical justification for our approach (Section 3.1) **[Suggested by gLKD]**
- Expanding empirical evidence through comprehensive experiments (Tables 2, 11, 12 and Figures 6, 7) **[Suggested by gLKD,FG4r,xTZD]**

**We have conducted extensive additional analyses**:
- Calibration and overconfidence analysis (Section 3.3) **[Suggested by gLKD,FG4r]**
- Confirmation bias investigation (Appendix A.6) **[Suggested by gLKD,FG4r]**
- Convergence analysis (Appendix A.7) **[Suggested by gLKD]**
- Logit complexity study (Appendix A.4) **[Suggested by FG4r]**

**We have broadened the scope of experiments by**:
- Evaluating different batch sizes in our ablation study **[Suggested by xTZD]**
- Including Large Language Model results (Appendix A.5) **[Suggested by xTZD]**
- Investigating various teacher model roles **[Suggested by FG4r]**

Furthermore, we have improved the paper's clarity by standardizing notations throughout and including standard deviations in our results. These revisions comprehensively address the reviewers' concerns while strengthening the paper's contributions.

---

### Decision · Action_Editor_Pza7 · 2025-02-12

**Recommendation:** Reject

**Comment:**

This paper introduces LumiNet, a knowledge distillation method tackling overconfidence, capacity gaps, and teacher-student fooling. Experiments on CIFAR-100, ImageNet, and MS COCO compare it with feature-based KD methods, alongside a theoretical analysis.

- Reviewer GLKd criticized the paper for lack of evidence to support the novelty and theoretical contribution claims. Despite discussions, the reviewer remains skeptical and recommends rejection.
- Reviewer xTZD raised concerns about missing ablation studies and language model experiments. Most were addressed, leading to a leaning accept rating.
- Reviewer FG4r highlighted clarity issues, overclaims on overconfidence, and missing discussions on teacher model selection. After revisions, they suggest minor updates and lean towards acceptance.

Given GLKd’s unresolved concerns, I recommend rejection but encourage resubmission after further revisions.

**Audience:**

I think some individuals in TMLR's audience would be interested in knowing the findings of this paper, after these issues are cleared.

**Claims And Evidence:**

Based on the reviewers' feedback, the claims in the submission are not fully supported by accurate, convincing, and clear evidence. The main issues are:

Unsubstantiated Novelty Claim – Reviewer GLKd argues that the novelty claim about LumiNet's proposed techniques (e.g., "Perception" and "LumiNet Loss") is unsubstantiated since the method resembles existing approaches like entropy regularization.

Unsubstantiated Theoretical Contribution Claims – Theoretical contributions are presented without sufficient empirical or mathematical support, particularly in Section 3.3.

Potential Plagiarism – The revised manuscript allegedly includes a convergence proof taken from an external source without proper citation.

False References – Some cited works do not support the corresponding claims, particularly regarding Expected Calibration Error (ECE) and Maximum Calibration Error (MCE).

Ambiguous Terminology – Key terms, such as Mutual Information and ECE/MCE, lack precise definitions, making it difficult to verify or reproduce results.

While some concerns were addressed in revisions, these fundamental issues undermine the credibility of the paper’s claims.

**Resubmission Of Major Revision:**

The authors may consider submitting a major revision at a later time.